# Parameterized Adverse Lens Corruptions to Probe Model Robustness to Optical Tolerances

**Kai Bäuerle**                                                    *kaibauerle@gmail.com*
*University of Mannheim, Germany*

**Patrick Müller**                                          *patrick.mueller@uni-siegen.de*
*University of Siegen and Center for Sensor Systems (ZESS), Siegen*

**Ivo Ihrke**                                                      *ivo.ihrke@uni-siegen.de*
*University of Siegen and Center for Sensor Systems (ZESS), Siegen*

**Margret Keuper**                                              *keuper@uni-mannheim.de*
*University of Mannheim and Max Planck Institute for Informatics, Saarland Informatics Campus, Germany*

**Reviewed on OpenReview:** *https://openreview.net/forum?id=a93BmQRNxC*

## Abstract

Deep neural networks excel at image classification on benchmarks like ImageNet, yet they remain vulnerable to adverse conditions, including environmental changes and sensor noise, such as lens blur or camera noise. Consequently, the study of these adverse noise corruptions has been extensive. At the same time, image blur, naturally introduced in optical systems, has been widely ignored as a threat to model robustness. In fact, Gaussian blur has even been considered as viable defense against adversarial attacks. In this work, we challenge the common perception of blur as a rather benign data corruption and study optics-driven, blur-based adversarial attacks. Specifically, we introduce Adverse Lens Corruption (ALC), an optics-driven robustness probe that, through adversarial optimization, identifies worst-case lens blurs by optimizing Zernike polynomial-based aberrations. Unlike traditional noise-based attacks, ALC provides a physically-motivated continuous search space. This enables the analysis of model robustness to optics-driven blur corruptions and complements existing noise and corruption benchmarks.

## 1 Introduction

Intensive studies of deep neural network (DNN) architectures He et al. (2016); Tan & Le (2019); Liu et al. (2022c); Oquab et al. (2023); Radford et al. (2021); Kirillov et al. (2023), further enhanced with Neural Architecture Search (NAS) Dosovitskiy et al. (2021); Liu et al. (2022a) and other advanced training schemes Touvron et al. (2021); Chen et al. (2023), have achieved remarkable classification and segmentation results. In the last decade, performances of models on popular benchmarks like ImageNet Russakovsky et al. (2015), CIFAR-100 Krizhevsky & Hinton (2009), and COCO Segmentation Lin et al. (2015) have been remarkably increasing. Yet, most models struggle when they are confronted with adverse conditions, *i.e.*, any change of light, weather, and other optical distortions Hendrycks & Dietterich (2019); Müller et al. (2023); Bäuerle et al. (2024) that deviate from the training conditions. Although DNNs can achieve high accuracy on in-distribution data, they usually suffer from a great performance degradation under such real-world distribution shifts Sakaridis et al. (2021); Hendrycks & Dietterich (2019). Thus, enormous research efforts are invested in the analysis Saikia et al. (2021); Hendrycks & Dietterich (2019); Kar et al. (2022) and improvement Hendrycks et al. (2020; 2021) of model robustness under different kinds of disturbances.

To understand the behavior of models under adverse conditions, there are several benchmarks and methods that allow for testing against adverse environmental and sensor-related conditions Sakaridis et al. (2021); Hendrycks & Dietterich (2019); Kar et al. (2022); Müller et al. (2023), and within adversarial settings Goodfellow et al. (2015); Croce et al. (2021); Madry et al. (2018). One important line of research models image corruptions such as weather changes, blur and noise as post-process effects in a controlled setting Dodge & Karam (2017); Hendrycks & Dietterich (2019); Kar et al. (2022); Müller et al. (2023), which includes benchmarks like 2D and 3D common corruptions and OpticsBench. These benchmarks are organized into a small number of discrete severity steps that allow to track the decline in accuracy with more difficult images and to establish where models fail in discriminating different categories. However, while these benchmarks are informative and provide a standardized way of testing, they do not provide the means for a fine-grained model-specific analysis comparable to adversarial attacks that find worst-case scenarios within clearly mathematically defined bounds Madry et al. (2018). Classical adversarial attacks Goodfellow et al. (2015);

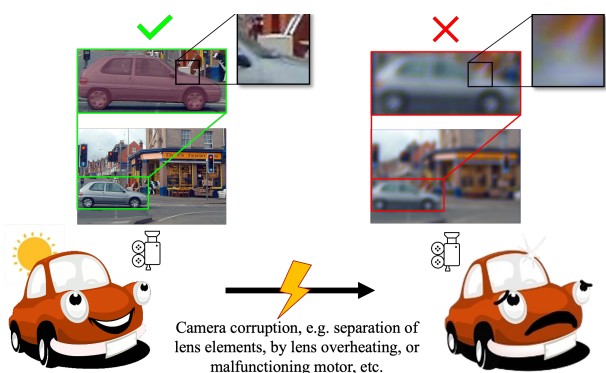

Figure 1: In safety critical applications such as autonomous driving, changes of the camera output can cause challenging data shifts that affect network performance, leading to potentially catastrophic failure. The figure shows, in a simplified fashion, how camera changes, *e.g.* due to conditions in the operational environment such as temperature and humidity changes, can result in incorrect detections. The root causes of such camera changes can be attributed to a separation or change of shape of lens elements, *e.g.*, by lens overheating or the malfunctioning of focusing actuators within the lens.

Croce et al. (2021) provide such clearly defined bounds on a point-wise level, understanding the image formation process for image $y$ as

$$y = h * x + n, \tag{1}$$

where the i.i.d. additive noise $n$ is adversarially optimized.

Complementing existing attacks, we introduce adverse lens corruption (ALC) designed to evaluate the robustness of image classification models against optical blur corruptions, *i.e.* we are optimizing the optical kernel $h$ that acts on the input signal $x$ via *convolution*. The proposed ALC learns a linear combination of primary optical aberration effects, such as coma, astigmatism and spherical aberration, to represent lens blur. For this, we expand the wave aberration into Zernike polynomials and optimize the coefficients with different constraints to produce a dataset and model-specific worst-case lens blur. The parameterization allows to analyze worst case blur in terms of orthogonal components such as plain defocus (very common), astigmatism, or trefoil.

We show that ALC yields model and dataset specific lens corruptions that effectively reduce the model accuracy. Compared to the worst-case OpticsBench Müller et al. (2023) corruption per model, ALC is stronger at a comparable corruption size. Compared to PGD for single image attacks, which uses an $\ell_{\text{inf}}$ constraint set, ALC provides complementary information, *e.g.* the main class-wise confounders are different.

While adversarial training is mostly used to defend against classical adversarial attacks such as Croce et al. (2021) and data augmentation is often not model-specific Hendrycks & Dietterich (2019), we also test an ALC-based adversarial training and demonstrate that adversarially trained models exhibit improved stability on CommonCorruptions Hendrycks & Dietterich (2019) and OpticsBench Müller et al. (2023).

Our main contributions are as follows:

- We propose adverse lens corruption (ALC), a white-box robustness probe based on an adversarial optimization, which finds the worst-case lens blur per dataset and model using gradient ascent on a linear combination of aberration basis vectors, complementary to existing attacks that optimize additive noise. ALC runs in two modes: a generic mode producing one kernel per dataset and model (Sec. 4.1 - 4.5) and an image-specific mode (Sec. 4.6).

- We demonstrate with several experiments across multiple datasets (ImageNet variants, MS COCO) and models (*e.g.* SwinV2, ResNet, CLIP, SAM) which specific points in Zernike space the models identify as worst-case.

- The parameterization in Zernike polynomials allows to analyze worst case blur in terms of orthogonal components that are physically plausible, such as plain *defocus*, *astigmatism*, or *trefoil*.

- We use ALC for adversarial training and demonstrate that ALC makes the models more robust against lens blur.

## 2  Related Work

Model robustness and stability have been addressed from various perspectives. In the following, we first give a brief introduction to data augmentation methods for training and measuring classification robustness against corruptions, then briefly summarize literature on model hardening via adversarial training. Our proposed method introduces an optical adversarial attack, which can be used to benchmark state-of-the-art image classification models, as well as harden these models via adversarial training. While previous adversarial attacks purely model point-wise noise corruptions, ALC models adversarial blur inspired by real optical aberrations.

**Data Augmentation and Knowledge Distillation** Data augmentation helps improve image classification robustness by simulating real-world variations. AugMix Hendrycks et al. (2020) enhances robustness to common corruptions, while Müller et al. (2023) use optical blur kernels to address optical aberrations. Other approaches involve learned augmentation policies Cubuk et al. (2018), feature map perturbations Hendrycks et al. (2021); Erichson et al. (2024), frequency modifications Yucel et al. (2023), or style adaptation using generative models Hong et al. (2021); Xue et al. (2023); Zhang & Agrawala (2023); Ho et al. (2020). Knowledge Distillation (KD) transfers robustness from a teacher model, improving adversarial Goldblum et al. (2020); Zi et al. (2021); Huang et al. (2023); Zhao et al. (2022) and out-of-distribution resilience Zhou et al. (2023). While highly effective, KD depends on the availability of large pre-trained models for the target domain and requires significant computation power.

**Adversarial Attacks** Adversarial attacks exploit neural network vulnerabilities by applying subtle perturbations on specific areas of the image that cause misclassification Goodfellow et al. (2015). Prior research has frequently approached image blur from a defensive perspective. Since standard adversarial attacks rely on high-frequency perturbations, simple input transformations such as Gaussian blur or spatial smoothing have been proposed to denoise inputs and mitigate attack effectiveness Guo et al. (2018); Dziugaite et al. (2016). *In contrast, our work demonstrates that worst-case optical blur is not a sanitizing mechanism, but a distinct threat model that degrades feature representation differently than additive noise.* Generally, the methods are categorized into white-box (usually first order) Goodfellow et al. (2015); Madry et al. (2018); Carlini & Wagner (2017); Kurakin et al. (2017); Croce & Hein (2020a) and zero-order Andriushchenko et al. (2020) attacks, depending on the accessibility of the model's weights to the attacker. Croce & Hein (2020b) propose to use an ensemble of a combination of white-box and zero-order attacks.

In Dong et al. (2020), superpixel-guided attention is used to target critical image regions. Our adversarial lens corruption leverages light-based perturbations to deceive computer vision models without changing digital inputs. Fang et al. (2023) contrasted invasive and non-invasive forms of attacks, emphasizing their viability in real-world scenarios. Other methods use structured illumination or Fourier-domain light modulation Gnanasambandam et al. (2021), modify the light by exploiting the rolling shutter effect to introduce adversarial light patterns that are invisible to humans Sayles et al. (2021), adversarial shadows Zhong et al. (2022) or reflections Wang et al. (2023). In addition, physical adversarial attacks Li et al. (2019) are also applicable in other domains with *e.g.* adversarially modified metals that threaten prohibited item detection for X-Ray-based sensors Liu et al. (2023). Crucially, these approaches primarily model an external threat, relying on external hardware (*e.g.* projectors, SLMs) or environmental scene changes to actively inject adversarial patterns into the sensor. In contrast to these works, the proposed ALC introduces an internal, systemic threat model. Rather than assuming active external light manipulation, ALC simulates the passive, physical degradation of the optical system itself by optimizing its internal parameters via Zernike polyno-

mials. Consequently, the proposed ALC is designed to probe models for their susceptibility to physically grounded, non-malicious optical blur corruptions.

While previous works such as Karmon et al. (2018); Brown et al. (2018); Cheng et al. (2024) have advanced physically-constrained attacks either through local noise or universal patches, they primarily focus on perturbing local regions in the image; ALC attacks the whole image to model degradations of the optics. In a similar spirit, Schmalfuss et al. (2023) propose realistic adversarial attacks to model adverse weather. They adversarially optimize rain particles and render the resulting scene. In contrast to standard adversarial attacks Goodfellow et al. (2015), such weather attacks are usually not $\ell_2$-bounded. Our ALC attack is also not $\ell_2$-bounded, but constrained in an auxiliary aberration space within which we adversarially optimize a realistic optical model.

With adversarial training, models can learn to defend against adversarial attacks Goodfellow et al. (2015); Engstrom et al. (2019); Madry et al. (2018); Carlini & Wagner (2017). At the same time, adversarial training can shift texture bias towards shape bias Gavrikov et al. (2023). Our experiments indicate that this is also true for ALC training (see Appendix G).

## 3 Optimizing Adverse Lens Corruptions

While existing research on adversarial attacks has focused on optimizing high-frequency perturbations, represented by the noise term $n$ in Eq. 1, we propose a novel blur-based threat model grounded in physical optics that effectively *attenuates* high-frequency content. Although $h$ allows for arbitrary kernels, we ensure a principled exploration of the kernel space, by defining our threat model using an orthogonal polynomial basis (Zernike polynomials Born et al. (1999)), which provides a mathematically rigorous representation of optical aberrations.

Optical aberrations are caused by imperfections of lenses, that distort the ideal image formation by characteristic blur $h$. To this end, we model the point spread function (PSF), which mimics the characteristic lens blur describing the spatial impulse response $h$ of a linear optical system on an image.

More specifically, the PSF describes, how a point source that emanates a spherical wave is being modified by a lens. A perfect lens converts the diverging spherical wave into a wave converging near the geometric image point. However, the limiting aperture and imperfections of a real lens cause the exiting wave from a lens to distort from its ideal leading to a *spread* of the point response according to the theory of diffraction and aberrations Goodman (2017).

In general, this PSF varies with wavelength, location, and distance Goodman (2017). However, we make several practical simplifications to this general setting: 1) the local PSF can be assumed to be approximately constant within a small portion of the image. In this work, we therefore assume that the small ImageNet images are regions from a larger image (several megapixels) and that the PSF can be treated as constant. 2) the depth-dependence of the PSF is negligible beyond the hyperfocal distance, we therefore assume imaging at optical infinity to enable a convolutional treatment of the lens aberration effects. With these simplifications, the PSF only depends on the wavelength.

Our adverse lens corruption generates a PSF for each color channel, which we then use as a convolution kernel $h$ to generate a corrupted image $y$ from the clean image $x$ via convolution, denoted by '$*$':

$$y = h * x. \tag{2}$$

This is in contrast to classical adversarial attacks Goodfellow et al. (2015) that assume $h$ to be fixed to a Dirac impulse and that optimize an additive noise $n$ as in Eq. 1. In order to acquire the kernel $h$ tailored to optical aberrations, we define $h$ using optical basis functions $Z_n^m$ describing the phase deviation $W_\lambda$ from an ideal spherical wavefront in the exit pupil plane; see Restrepo et al. (2016) for a detailed treatment of the relation between wave and ray aberrations. To convert this phase deviation $W_\lambda$ to a PSF kernel $h$, we need to propagate the light wave to the image space. To achieve this, we use the linear system model for

diffraction and aberration as in Goodman (2017):

$$h_\lambda(u,v) = \left| \mathcal{F}\left\{ \mathrm{Circ}(\omega_u,\omega_v) \cdot e^{-j\frac{2\pi}{\lambda z}W_\lambda(\omega_u,\omega_v)} \right\} \right|^2 , \tag{3}$$

with $j$ being the imaginary number and $W_\lambda$ the wave aberration for wavelength $\lambda$ and $h_\lambda$ being the spatial impulse response, the PSF. For a circular exit pupil $Circ$, the wavefront is propagated from the exit pupil with coordinates $(\omega_u, \omega_v)$ to the image space at point $(u, v)$ at distance $z$ via the squared Fourier transform $\mathcal{F}$, to yield an incoherent PSF $h$ for wavelength $\lambda$ Goodman (2017). Physically, this operation transforms the phase-domain wave aberrations into an image-domain blur pattern, ensuring that $h$ is a physically-valid, non-negative convolution kernel. The squared Fourier transform propagates the pupil phase error $W_\lambda$ to the image plane, and the modulus-square reflects that the sensor measures intensity rather than amplitude. The resulting kernel is the PSF the lens would produce.

In Eq. 3, we expand the wave aberration $W_\lambda(\omega_u, \omega_v)$ into the orthogonal Zernike polynomials Born et al. (1999) denoted by $Z$, which allows for a parameterization of individual aberrations defined in multiples of wavelength $\lambda$:

$$W_\lambda(\omega_u, \omega_v) = \lambda \cdot \sum_{n,m} A_n^m(\lambda) \cdot Z_n^m(\omega_u, \omega_v) \tag{4}$$

Eq. 4 represents the wave aberration as a weighted sum of optical basis functions $Z_n^m$. The coefficients $A_n^m$ determine the strength and sign of each aberration mode, while Eq. 3 converts the resulting pupil-plane phase error into the image-plane PSF used as the convolution kernel. Each optical basis function $Z_n^m$ represents a meaningful physical representation of specific types of lens aberrations, such as astigmatism and coma. Our optical attack learns the coefficients $A_n^m(\lambda)$ depending on $\lambda$, *i.e.* for each color channel separately, to form a worst-case combination of optical aberrations for, *e.g.*, an image classification model.

**Adverse Lens Corruption**  Using the optical kernel from Eq. 3, we can corrupt the clean image $x$ with a convolution with the optical kernel $h_\lambda$. We can further scale the blur contribution via a corruption size $\nu$, which results in a corrupted image $adv_x^t$ at time step $t$,

$$adv_x^t = adv_x^{t-1} + \nu \cdot (x * h_\lambda^{t-1} - adv_x^{t-1}), \quad \nu \in [0,1], \tag{5}$$

where $h_\lambda^{t-1}$ denotes the optimized lens corruption at timestep $t-1$ of a gradient ascent based optimization and $adv_x^0 = x$, *i.e.* $h^0$ is the Dirac impulse. In every iteration, $adv_x$ will be clamped to the valid image range. Note that in our default setting, $\nu$ will be set to one such that the corruption is purely convolutional as in Eq. 2. However, this formulation with $\nu < 1$ also allows ALC to be turned into a classical additive attack. In this case, the corruption can be $\ell_2$ bounded as in Madry et al. (2018).

To use this process in an adversarial technique, we use our model parameters $\theta$ and the target $label_x$ to calculate the model loss $L$ (typically cross entropy loss) and update subsequently $A_n^m$ to increase $L$ in an iterative manner

$$A_n^{m,t} = A_n^{m,t-1} + \alpha \nabla_{A_n^{m,t-1}} L(\theta, adv_x^{t-1}, label_x), \tag{6}$$

where the coefficients $A_n^{m,t}$ determine the optical kernel $h_\lambda^t$ from Eq. 5 for all three colors $\lambda$ as in Eqs. 3 and 4.

**Constraints on the Magnitude of ALC**  We restrict the model to 8 out of the first twelve Zernike Fringe modes, which can be mixed into one 3D kernel $(u, v, \lambda)$. Table 1 highlights the selected modes in bold.

Next to the limitation of the selected modes, the adversarial attack is restricted by the magnitude of the corruption coefficients $A_n^m$, via normalizing for each color channel $k_\lambda \in \{r, g, b\}$ if the overall corruption size is larger than a threshold $\tau$:

$$A_{n,k_\lambda}^m = \begin{cases} \frac{\tau \cdot A_{n,k_\lambda}^m}{\sum_{n,m}|A_{n,k_\lambda}^m|}, & \text{if } \sum_{n,m}|A_{n,k_\lambda}^m| > \tau \\ A_{n,k_\lambda}^m, & \text{otherwise.} \end{cases} \tag{7}$$

| # | Name | # | Name |
|---|------|---|------|
| 1 | piston | 7 | **horizontal coma** |
| 2 | tilt u | 8 | **vertical coma** |
| 3 | tilt v | 9 | **spherical** |
| 4 | **defocus** | 10 | **oblique trefoil** |
| 5 | **oblique astigmatism** | 11 | **vertical trefoil** |
| 6 | **vertical astigmatism** | 12 | sec. vert. astigmatism |

Table 1: First twelve Zernike Fringe modes. We select numbers 4-11 for our optical attack (bold). The exclusion of modes 1-3 is due to optical considerations: the piston term does not cause a kernel intensity change and modes 2 and 3 only cause an image shift without blur. We therefore exclude these modes from our analysis. The remaining selected modes are all Zernike primary aberrations, including primary trefoil (modes 10 and 11). A visualization is given in the appendix in Fig. 9.

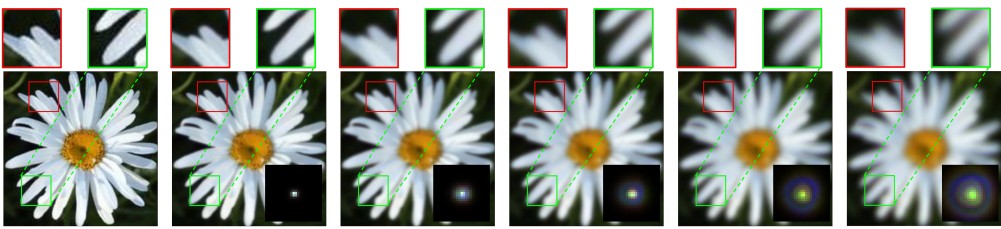

Figure 2: Adversarial optical attack examples with increasing thresholds $\tau$. The image was created with an optical kernel from the PSF attack with the ResNet50. From left to right, $\tau$ increases from 0 to 5, where each inlet shows the corresponding kernel.

Similar to the restriction of the combined magnitude of the corruption, we also restrict each coefficient to a given element-wise threshold $\tau_e$ via:

$$A_{n,k_\lambda}^m = \begin{cases} \text{sign}_{A_{n,k_\lambda}^m} \cdot \tau_e, & \text{if } |A_{n,k_\lambda}^m| > \tau_e \\ A_{n,k_\lambda}^m & \text{otherwise,} \end{cases} \tag{8}$$

where $\text{sign}_{A_{n,k_\lambda}^m}$ denotes the sign of the coefficient.

This restriction implements a fair distribution over the corresponding corruptions. The resulting kernel is then $\ell_1$-normalized per color-channel to avoid any brightness changes, when applied to an image. An example of an optically corrupted image with mixed corruption can be examined in Figure 2 for increasing corruption budget $\tau$.

## 4 Experimental Evaluation

In the following, we evaluate different computer vision models with ALC. We use ALC in two modes: a generic, dataset-level mode that produces a single worst-case kernel per model and dataset, and an image-specific mode that optimizes a separate kernel per image.

We structure the evaluation from probe to attack to defense and begin with the generic mode, for a simple physical reason: a real lens resides in one fixed aberration state that affects all captured images alike. The generic mode mirrors exactly this setting with one optimized kernel per model and dataset and therefore characterizes a model's structural sensitivity to a degraded optical system, which is the question raised by optical tolerances. This probe setting underlies the model comparison (Sec. 4.1), the study of dataset size and complexity (Sec. 4.2), the transfer experiments (Sec. 4.3), the ablation of the aberration space (Sec. 4.4), and the segmentation experiments (Sec. 4.5). Only then do we switch to the image-specific mode (Sec. 4.6), in which ALC acts as a per-image adversarial attack: it no longer corresponds to a single physical lens state, but provides an upper bound on the achievable optical degradation and reveals which images and

classes are particularly vulnerable. Finally, Sec. 4.7 closes the loop and uses the corruptions identified by ALC for adversarial training.

**Implementation of the two modes** In the generic mode, ALC is optimized over the whole dataset and generates one data- and model-specific kernel. This kernel combines the worst-case corruption by iterating through the whole dataset and updating the coefficients $A_n^m$ that determine the optical kernel $h_\lambda^t$ as described in Eq. 5.

To achieve a fairly distributed updating process over all images, we iterate through the dataset in whole epochs: in each epoch, the coefficients $A_n^m$ are updated only on the training subset and subsequently evaluated on the validation subset. $A_n^m$ is updated via stochastic gradient descent with an initial learning rate of 0.001, a momentum of 0.9, and a weight decay of 0.0001. The coefficients show no major further adjustments after roughly 1M image presentations. We therefore run the probe for one epoch on the ImageNet21k and ImageNet1k, for 10 epochs on the ImageNet100, and for 100 epochs on the ImageNette dataset, if not stated differently.

The image-specific mode is not conducted over the whole dataset at once, instead it iterates over each image multiple times (40 in our experiments) and updates the coefficients $A_n^m$ to obtain an image-specific adversarial kernel $h_\lambda^t$. After each image, the coefficients are re-initialized randomly in the range of $[-0.1, 0.1]$, which ensures that the optimization starts near a diffraction-limited state and explores the local aberration space starting from an ideal lens.

The DNNs we evaluate are trained on different publicly available subsets of ImageNet Russakovsky et al. (2015) to allow for extensive experiments and investigate the behavior of the probe on datasets with different complexity, *i.e.* ImageNette Howard (2023), ImageNet-100 Tian et al. (2020), ImageNet-1k Russakovsky et al. (2015), and ImageNet-21k Russakovsky et al. (2015). To complement image classification, we also conduct experiments on image segmentation on the COCO segmentation dataset Lin et al. (2015) using SAM Kirillov et al. (2023). All experiments are conducted with $\tau$ and $\tau_e$ values of 4 to provide compatibility to OpticsBench Müller et al. (2023). We also report an ablation of how $\tau$ affects the accuracy in the Appendix in Section B.

## 4.1 Model comparison

To evaluate the generic ALC probe, we train multiple models on multiple datasets. Subsequently, the optical kernel for the specific dataset - model combination is trained and evaluated (see Appendix A.1 for details). We evaluated on CNN-based models, such as ResNet50 He et al. (2016), AlexNet Krizhevsky & Hinton (2009), DenseNet161 Huang et al. (2017), EfficientNet b0 and b4 Tan & Le (2020), VGG16 Simonyan & Zisserman (2015), MobileNetV3 large Howard et al. (2019), ConvNeXt Liu et al. (2022b) and ConvNeXt v2 Woo et al. (2023), as well as on transformer-based models like Vision Transformer Dosovitskiy et al. (2021) and Swin Transformer v2 Liu et al. (2022a). Furthermore, we probe state-of-the-art foundation models, such as CLIP Radford et al. (2021), DINO v2 Oquab et al. (2023) and SAM Kirillov et al. (2023). Note that for this experiment, we optimize a single adverse kernel per model and dataset. Testing the model robustness at inference time is thus very efficient.

Table 2 shows the classification accuracy of different pretrained models on ImageNet1k under clean and ALC attacked conditions. While all evaluated models perform poorly, ViT (base) is one of the most robust models against the optical attack. The model achieves a relatively high corrupted accuracy of 0.177, followed by EfficientNet-b4 (0.149). Both of these models are also able to classify best on the clean dataset. Yet, all models see a substantial decrease in accuracy upon being attacked by the optical attack, with the SwinV2-tiny seeing the highest accuracy decrease at 0.736, whereas MobileNet v3 sees the lowest absolute accuracy while being attacked.

Figure 3 shows the generated kernels of three of the models, from Table 2. The corruptions are displayed via the generated kernels both combined and per color channel. Even though the robustness of the Vision Transformer-base model (bottom) is higher than the robustness of the SwinV2-tiny and the MobileNetV3-large models, the generated kernels of the red and green channel are quite similar. The adverse lens corruption chooses, for all of these models, a similar corruption combination for those two channels. However, the

|  | Clean ↑ | OpticsBench ↑ | ALC ↑ | Δ | $\ell_2$ |
|---|---|---|---|---|---|
| **ResNet50** | 0.809 | 0.130 | 0.093 | 0.716 | 44.618 |
| **VGG16** | 0.716 | 0.044 | 0.052 | 0.664 | 40.623 |
| **DenseNet161** | 0.772 | 0.141 | 0.074 | 0.698 | 46.822 |
| **EfficientNet b0** | 0.777 | 0.116 | 0.049 | 0.728 | 45.738 |
| **EfficientNet b4** | 0.834 | 0.127 | 0.149 | 0.685 | 47.264 |
| **MobileNet v3** | 0.753 | 0.072 | 0.029 | 0.724 | 43.500 |
| **ViT (base)** | **0.853** | **0.246** | **0.177** | 0.676 | 43.478 |
| **SwinV2-tiny** | 0.821 | 0.132 | 0.085 | **0.736** | 46.002 |
| **SwinV2-small\*** | 0.837 | 0.183 | 0.119 | 0.718 | 45.014 |
| **CLIP** | 0.761 | 0.104 | 0.078 | 0.683 | 48.274 |
| Σ | - | - | - | - | 45.13 |

Table 2: Accuracy for different pretrained classification models, including clean accuracy, accuracy for our ALC probe and the difference (Δ), evaluated on ImageNet1k using several pretrained models. *=Multiple Seeds. For comparison, we also report the accuracy for the hardest corruption on OpticsBench Müller et al. (2023) per model for severity 5, where the $\ell_2$-distance is 46.0 for OpticsBench. The average $\ell_2$-distance across OpticsBench corruptions of 44.32 is comparable to ALC with an average 45.13 for $\tau = 4.0$. The single optimized ALC corruption per dataset is stronger than the OpticsBench corruption at a comparable corruption level.

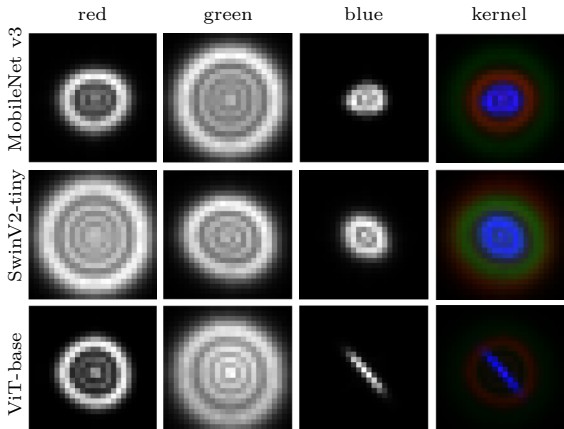

Figure 3: Generated kernels with ALC for ImageNet1k Russakovsky et al. (2015) and 3 different models. For MobileNet v3 large (top), the SwinV2-tiny (middle), and the ViT-base (bottom), the plot shows the generated kernel for each color channel (red, green, and blue, normalized to one for displaying purposes) and all channels combined.

corruptions in the blue channel differ significantly. The Vision Transformer adversarial blue kernel mostly shows vertical astigmatism, the adversarial blue kernel of the other two models are a combination of defocus, oblique astigmatism, and spherical. This might indicate that the adversarial kernel for ViT-base was harder to optimize - defocus does not seem to be optimal in this case. ViTs have been shown to be less texture-biased than e.g. ResNets Hermann et al. (2020). We hypothesize that texture-biased models suffer more severely under blur corruptions, because blur can remove texture details.

We also compare our results for the ALC attack with OpticsBench Müller et al. (2023) for severity 5, which gives a similar $\ell_2$-distance (on average 44.32 for OpticsBench versus 45.13 for ALC). Since ALC finds the worst-case lens blur, we compare to the hardest OpticsBench optical corruption at severity 5 for each model. The $\ell_2$-distance for the hardest OpticsBench corruption is on average 46.0, marginally larger than for ALC. At the same time, the accuracies for ALC are significantly lower than for OpticsBench, which underlines the efficacy of our method to find within a comparable $\ell_2$-budget a model-specific *worst-case lens blur*. Training a ResNet50 with OpticsAugment Müller et al. (2023) improves robustness against ALC by +5.4% on ImageNet100. However, ALC finds worse corruptions compared to OpticsBench. All generated kernels from the optical experiments conducted for Table 2 are shown in Section C (Appendix).

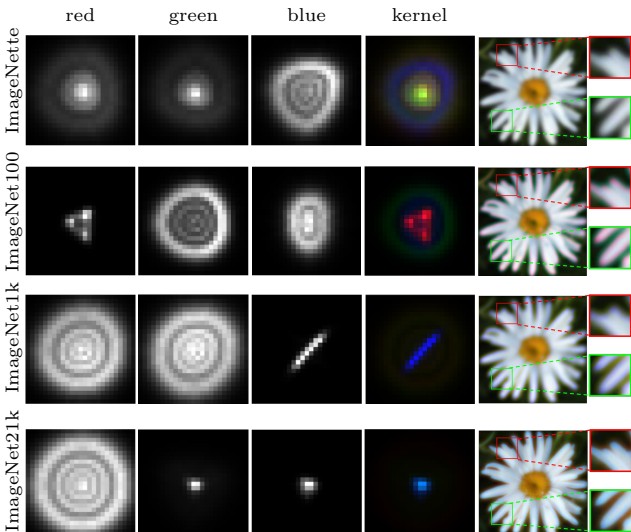

Figure 4: Difference in optical kernel generation for ImageNette, ImageNet100, ImageNet1k and ImageNet21k. The kernels for individual color channels are normalized to one for better visibility.

| Model | ImageNette | ImageNet100 | ImageNet1k | ImageNet21k |
|---|---|---|---|---|
| ImageNette | 0.299 | 0.460 | 0.337 | 0.485 |
| ImageNet100 | 0.237 | 0.158 | 0.224 | 0.318 |

Table 3: Accuracy of ResNet50 models under generic ALC kernels optimized on a *different* dataset. Columns: datasets on which the ALC kernel was optimized. Rows: dataset used for evaluation.

### 4.2 ImageNet Subsets

To study the effect of ALC on dataset size and complexity, we also evaluate on different image datasets. To differentiate these datasets in size and complexity, we used subsets of the ImageNet dataset, such as the ImageNette dataset Howard (2023), the ImageNet100 dataset Tian et al. (2020), the ImageNet1k dataset Russakovsky et al. (2015) and the whole ImageNet21k dataset Russakovsky et al. (2015).

Figure 4 compares the optical kernels found by the generic ALC probe with the same model architecture for the four different ImageNet subsets. The number of images and classes increases from top to bottom by row. Where in the top row only 10 classes are present, while adversarially attacking the ResNet50 model, in the bottom row, there are over 21k classes. However, not only the complexity, also the number of images in the dataset increases. In the figure, the first three columns show the generated optical kernel for each color channel. The fourth column displays the combined kernel, and the last column presents an example image perturbed by the attack. Figure 4 and the evaluation results in Subsection 4.1 indicate that the adverse lens corruptions are model and dataset specific. Figure 5 shows the mean magnitudes and variances for the learned aberration coefficients for ResNet50. It confirms that the variance across different ALC seeds is low while differences across datasets are significant. Figure 13 in the appendix shows that the variance increases for models trained with different seeds, while the adversarial blur corruption remains characteristic.

### 4.3 Transfer Attack

To evaluate to which extent the resulting corruption combinations found by the generic probe in Subsections 4.1 and 4.2 transfer to other model architectures or datasets, we conducted a set of transfer experiments in which the fixed kernels are applied as static corruptions. Specifically, we take the kernels obtained by the generic probe with ResNet50 on the four ImageNet subsets and evaluate ResNet50 models on ImageNette

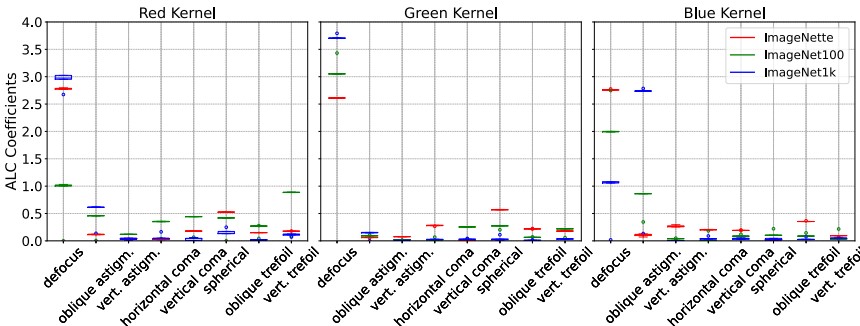

Figure 5: Comparison of the resulting ALC coefficients ResNet50 on multiple ImageNet subsets (ImageNette, ImageNet100, and ImageNet1k). See Figure 12 in the Appendix for a corresponding plot of signed ALC coefficients. The coefficient variances are low, while their respective differences are significant.

| Model | Dataset | Clean | full ALC | No Defocus | No Spherical |
|---|---|---|---|---|---|
| ResNet50 | ImageNette | 0.795 | 0.299 | 0.341 | 0.531 |
| ResNet50 | ImageNet100 | 0.801 | 0.109 | 0.230 | 0.441 |
| ResNet50 | | 0.809 | 0.102 | 0.267 | 0.564 |
| EfficientNet b4 | ImageNet1k | 0.834 | 0.149 | 0.374 | 0.619 |
| CLIP | | 0.761 | 0.078 | 0.148 | 0.410 |

Table 5: Ablation of ALC parameter space on ResNet50 He et al. (2016), EfficientNet b4 Tan & Le (2020) and CLIP Radford et al. (2021). We subsequently remove the dominant (largest) aberrations. First row: ResNet50 trained and evaluated on ImageNette Howard (2023). Second row: ResNet50 on ImageNet100. Third to fifth row: ResNet50, EfficientNet, and CLIP correspondingly trained on the ImageNet1k dataset Russakovsky et al. (2015).

and ImageNet100 under these fixed kernels. Results in Table 3 indicate the dataset specificity of ALC. Corruption combinations from the same dataset are more efficient than corruptions from different datasets.

| Attack | CLIP | ConvNeXtV2 | DINOV2 |
|---|---|---|---|
| – | 0.761 | 0.923 | 0.912 |
| ALC Attack | 0.303 | 0.540 | 0.670 |

Table 4: Accuracy of state-of-the-art models (CLIP, ConvNeXtV2, DINOV2) with and without the corruption combination from a ResNet50 ALC attack on the same dataset.

To further investigate the generalizability of the ALC attack, we evaluate the performance of computation expensive state-of-the-art foundation models against a corruption combination from ResNet50 on the same dataset (ImageNet1k). Specifically, we apply these optical perturbations to three diverse architectures: CLIP, ConvNextV2, and DINOv2. The results are summarized in Table 4. ALC, originally crafted using ResNet50, remains effective across model design. However, the model specific ALC attack is reducing the accuracy of certain models, such as CLIP, significantly more, *i.e.* to 0.078 vs. 0.303 (refer to Table 2).

## 4.4 Corruption Ablation Study

In the previous subsections (Subsection 4.1 & 4.2), we showed, that adverse lens corruptions introduce some defocus into the adversarial kernel. In this subsection, we elaborate, which alternative corruptions are getting increasingly important, while defocus, which has been shown in Table 5 to be the dominating adverse aberration, is not available. We justify this choice by noting that defocus can be corrected by changing the position of the image sensor. Therefore, we dismiss the defocus corruption as an additional restriction and rerun the dataset-level probe from Subsection 4.1. The result of this restricted probe can be examined in Table 5 with three different models (Resnet50, EfficientNet & CLIP) and three ImageNet subsets (ImageNette, ImageNet100 & ImageNet1k).

After defocus, the most prominent aberration is spherical aberration, which is caused by spherical lenses. Spherical lenses are used in most systems, because of the simple manufacturing process. Spherical aberration can also be corrected using aspherical lenses, which are *e.g.* common in mobile plastic lenses, but are often

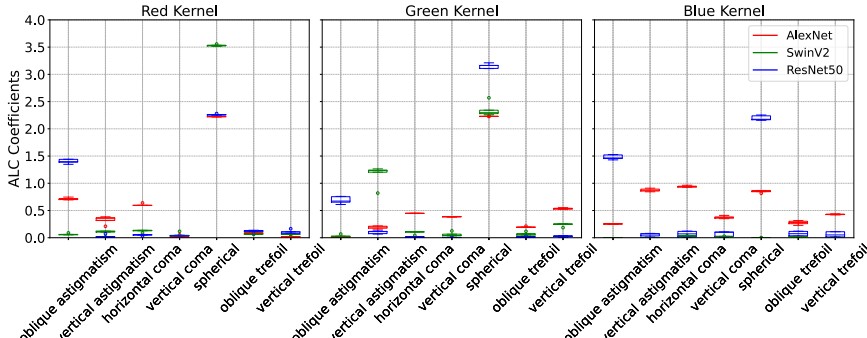

Figure 6: Comparison of the absolute coefficient values $A_n^m$ without defocus on three different ImageNet subsets (ImageNette, ImageNet100 and ImageNet1k). ALC was applied to the same model architecture (ResNet50). The models are trained with the same seed, however ALC was evaluated on 5 different seeds per dataset.

| Model | Attack | AP50 | AP75 |
|---|---|---|---|
| SAM base | - | 0.405 | 0.241 |
| | ALC | 0.198 | 0.102 |
| SAM huge | - | 0.818 | 0.460 |
| | ALC | 0.621 | 0.267 |

Table 6: Segmentation results on the COCO validation dataset. Evaluated on pre-trained SAM base Kirillov et al. (2023) and SAM huge Kirillov et al. (2023). ALC significantly reduces the Average Precision (AP).

costly to manufacture. However, the last column shows that correcting for spherical aberration would lead to significantly higher accuracies.

Figure 6 shows that the learned kernels are equally dataset characteristic when defocus is removed as an aberration. See also Figures 18 and 19 in the Appendix.

## 4.5 Segmentation Experiments

To evaluate the effectiveness of adverse lens corruptions in a segmentation task, we used two SAM (Segment Anything Model) models Kirillov et al. (2023). Specifically, we evaluate on the COCO Segmentation Dataset Lin et al. (2015) to assess whether the ALC could compromise the performance of the SAM model. In Table 6, the two SAM models (base & huge) are evaluated in terms of AP50 and AP75 prior and post adverse lens corruptions. Especially the SAM base variant, with a significant drop of 50% in comparison to the clean data, is vulnerable to blur corruptions. The SAM huge variant is quite robust against ALC when comparing AP50. However, on a more fine granular level, the performance on AP75 drops drastically. We can conclude that the rough shapes can still be well segmented by SAM huge while optical aberrations blur the image data such that the segmentation becomes incorrect on fine details.

## 4.6 ALC Attack per Image

In the previous subsections (Subsections 4.1 to 4.5), we used the ALC attack as a dataset-level robustness probe, *i.e.*, to generate a single mixed corruption kernel for the entire dataset. Here, we consider an image-specific variant of ALC. Instead of iterating over the whole dataset and updating the kernel after each batch, the attack iterates over one image multiple times (max. 40) and updates the kernel. This altered approach provides us insights into image-specific corruptions.

In comparison to the previous approach, the kernel and corruption diversity is increased significantly, as some images are more vulnerable against different corruptions. As a result, the accuracy between the classes also differs significantly. Table 7 in the appendix shows the drop in accuracy for each ImageNette class for the image-specific ALC on ResNet50, SwinV2, and AlexNet. Some classes, such as the *golf ball* class, have

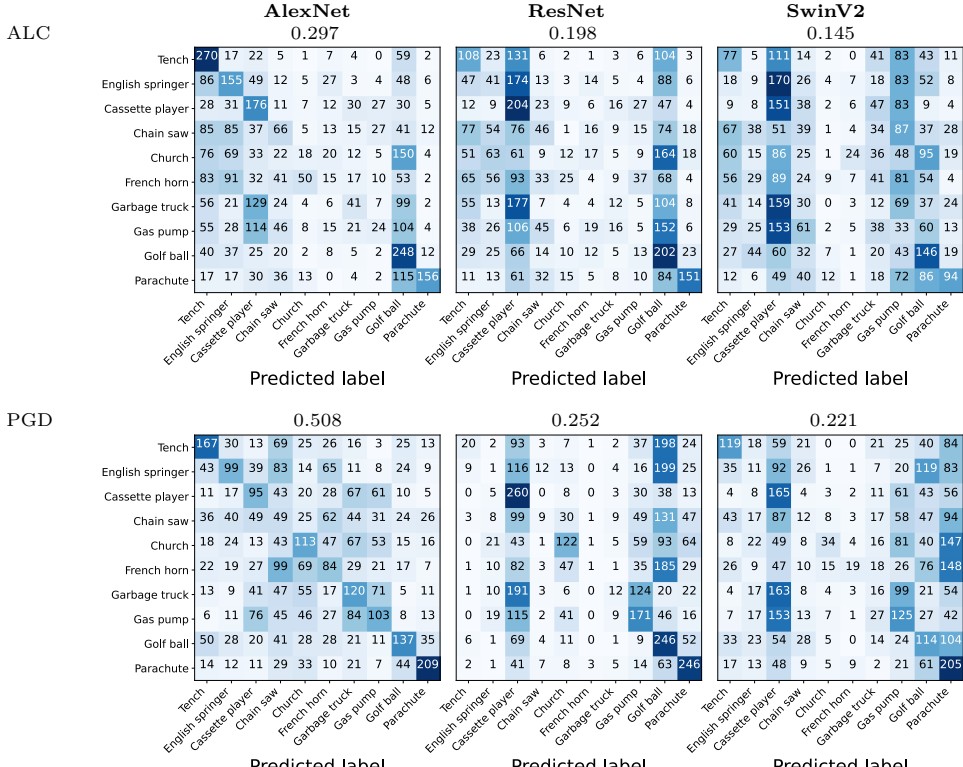

Figure 7: Confusion matrices for different models attacked by our proposed Adversarial Lens Corruption (ALC) (top) and PGD (bottom) evaluated on the ImageNette dataset. The PGD attack was conducted with an $\epsilon$ bound of $\frac{2}{255}$. Each confusion matrix originates from a different model - from left to right: AlexNet, ResNet and SwinV2. The accuracies of the attacked models are displayed on top of each confusion matrix. While the absolute accuracies between the differently bounded attacks can not be directly compared, it is interesting to see that different classes are most confused by the different nature of attack (blur versus noise).

significantly lower accuracy drops while being attacked by ALC than other classes (e.g. *church* class), which holds for all tested models. A detailed corruption distribution over each class is shown in the appendix in Section E, as well as an accuracy drop comparison for ImageNet1k.

Other adversarial attack methods, such as PGD Madry et al. (2018) and FGSM Goodfellow et al. (2015), attack the input images on a pixel-level and can not be reproduced by creating attack-based camera lenses. To compare the pixel-level attacks against our optical attack, we attacked the same models as in Figure 7 with an $\ell_{\inf}$ bounded 40 step (same as for ours) PGD attack with $\epsilon = \frac{2}{255}$. Because ALC is constrained in Zernike-coefficient space rather than by an $\ell_p$ budget (Sec. H), the two attacks are not matched in magnitude. We therefore compare only the pattern of confusions they introduce, which differs significantly across classes.

## 4.7 Adversarial Training

After finding the most harmful optical kernel for a dataset-model combination in Subsections 4.1 & 4.2 and the image-model combination in Section 4.6, in this subsection, we adversarially train the models to increase the robustness against optical corruptions. For a stable training process, we conducted multiple training processes, varying the restriction values $\tau$, as well as delaying the adversarial part of the training by varying the number of epochs, and fine-tuning with pretrained models via the optical attack from Section 3. More details and the result of these different approaches can be examined in the Appendix in Section F.

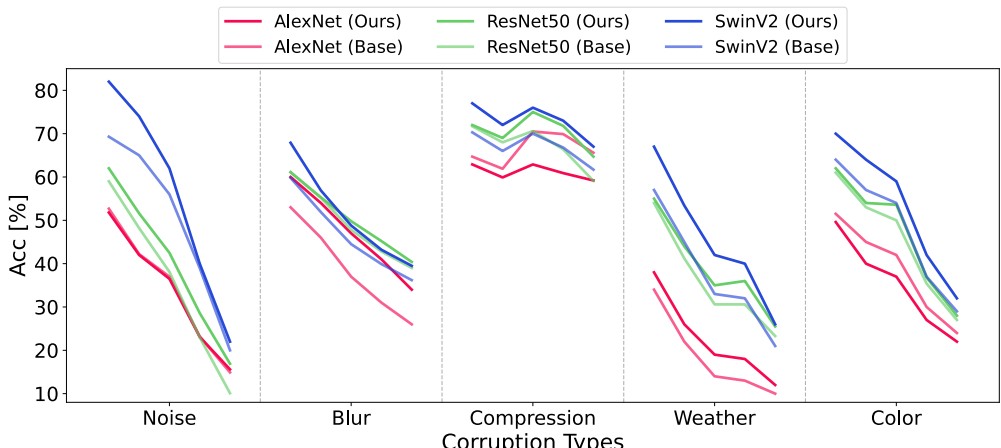

Figure 8: Adversarially trained models with ALC (ours) compared to baselines (Base). The models are evaluated on ImageNette for 2D Common Corruptions Hendrycks & Dietterich (2019) and OpticsBench Müller et al. (2023) corruptions.

Figure 8 shows AlexNet, ResNet50 and SwinV2 adversarially trained models against the models trained only on clean data. These models are evaluated on the Common Corruptions Hendrycks & Dietterich (2019) and OpticsBench Müller et al. (2023) dataset, divided into five sections (Noise, Blur, Compression, Weather, Color) and five severities (left to right). All three models, adversarially trained, outperform the respective baseline over all sections and severities marginally. In particular, it is interesting to observe that adversarial training on blur corruptions can improve the model behavior on noise corruptions.

## 5 Discussion

Our experiments demonstrate that optimizing $h$ via Zernike polynomials reveals model vulnerabilities that pixel-wise noise attacks miss. While additive noise typically introduces high-frequency perturbations, ALC acts as an adversarial low-pass filter. This distinction is crucial for a holistic robustness analysis: a model robust to pixel noise (e.g., via standard adversarial training) is not necessarily robust to optical aberrations. **This fundamental difference in the frequency domain explains why ALC addresses a dimension of robustness not covered by established methods listed in benchmarks like RobustBench**. Most standardized attacks (e.g., PGD or AutoAttack) focus on $\ell_p$-bounded high-frequency noise. In contrast, ALC probes the model's sensitivity to structured, low-frequency degradations inherent to the physical image formation process. Consequently, ALC should be viewed as a complementary diagnostic tool and training regularizer rather than a direct competitor to pixel-budget attacks. It quantifies shape bias by acting as a differentiable low-pass filter, revealing that texture-biased architectures are more vulnerable than shape-biased ones, as further detailed in Appendix Section G.

By parameterizing the adversarial search space via Zernike polynomials, we ensure that the generated kernels are built from realistic optical aberrations rather than arbitrary mathematical perturbations. Unlike standard pixel-noise attacks, our constraints restrict the search to the set of feasible optical aberrations rather than allowing arbitrary pixel manipulation. That said, a worst-case point in the continuous Zernike space is not guaranteed to correspond to a single manufacturable lens. Practical aspects are discussed in Appendix I.

## 6 Conclusion

Adversarial lens corruption (ALC) probes the robustness of models to lens blur, which complements standard noise-based adversarial attacks Goodfellow et al. (2015). Unlike prior work that utilizes blur as a defense, we formulate lens blur as an overlooked adversarial threat. By optimizing the linear combination of Zernike Polynomials, ALC identifies the aberrations that contribute most to the worst-case lens blur. We demonstrate

with a large number of experiments, that ALC finds a model- and dataset-specific lens blur. When used for adversarial training, ALC increases model robustness on Common Corruptions as well as on blur corruptions.

## Broader Impact Statement

While ALC demonstrates a method to intentionally degrade model performance via optical blur, its primary purpose is defensive. We acknowledge the dual-use potential for stress-testing real imaging pipelines but emphasize that our findings facilitate the development of robust training schemes against non-malicious environmental corruptions.

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
