This supplementary material provides detailed information on models, datasets and implementation, and additional experimental results. It is structured as follows:

- Dataset and Implementation details in Section A

- Ablation on threshold parameters of ALC in Section B

- Kernel visualizations in Section C

- Ablation on the optical corruptions in Section D

- Image-specific kernels and per-class results in Section E

- Implementation details for adversarial training with ALC for the experiments in Section 4.7 of the main paper are reported in Section F

- Discussing the Shape-Texture bias in the Context of ALC in Section G

- Discussing the non-$\ell_2$-boundedness of ALC in Section H

- Discussing practical aspects for lens and system designers in Section I

## A  Datasets and Implementation Details

To evaluate the robustness of multiple image classification models, subsets of the ImageNet Russakovsky et al. (2015) dataset are used. The ALC attack, in Section 3, uses the data and implementation process with the details from Subsection A.1 & A.2.

### A.1  Implementation Details

The generic ALC attack, which is trained over the whole dataset, generates one data and model specific kernel. This kernel combines the worst case corruptions by iterating through the whole dataset and updating the coefficients $A_n^m$ to determine the optical kernel $h_\lambda^t$ as described in Eq. 5. A visualization of kernels for single coefficients $A_n^m$ without mixing and the effect of mixing with defocus is provided in Fig. 9. To achieve a fair distributed updating process for each image, we iterate through the dataset in whole epochs. In each epoch, the coefficients $A_n^m$ of the ALC attack are only updated with the training subset and subsequently evaluated on the validation subset. $A_n^m$ is updated via stochastic gradient descent with an initial learning rate of 0.001, a momentum of 0.9, and a weight decay of 0.0001. The ALC attack has no major adjustments on the coefficients in less than 1M images. Thus, we attack the models for one epoch on the ImageNet21k dataset, one epoch on the ImageNet1k, 10 epochs on the ImageNet100 dataset, and 100 epochs on the ImageNette dataset, if not stated differently.

The image specific ALC attack is not conducted over the whole dataset at once, rather it iterates over each image multiple time to generate an image specific adversarial kernel $(h_\lambda^t)$. In our experiment, we iterate over each image 40 times and update the coefficients $(A_n^m)$ to determine the optical kernel $(h_\lambda^t)$ accordingly. After each image, the coefficients are initialized, again randomly, in the range of $[-0.1, 0.1]$. This initialization ensures that the optimization starts near a diffraction-limited state, exploring the local aberration space for the most impactful physical aberrations starting from an ideal lens.

### A.2  Datasets

In Section 4, we evaluate different computer vision models against our proposed, optical adversarial attack. The DNNs we evaluate are trained on different publicly available subsets of ImageNet Russakovsky et al. (2015) to allow for extensive experiments and investigate the attack behavior on datasets with different complexity, *i.e.*. ImageNette Howard (2023) is a dataset consisting of 10 ImageNet classes. It has 9,469 training and 3,925 validation images Howard (2023). ImageNet-100 Tian et al. (2020) uses 100 ImageNet classes with a total of 128k training and 5,000 validation images Tian et al. (2020). The ImageNet-1k

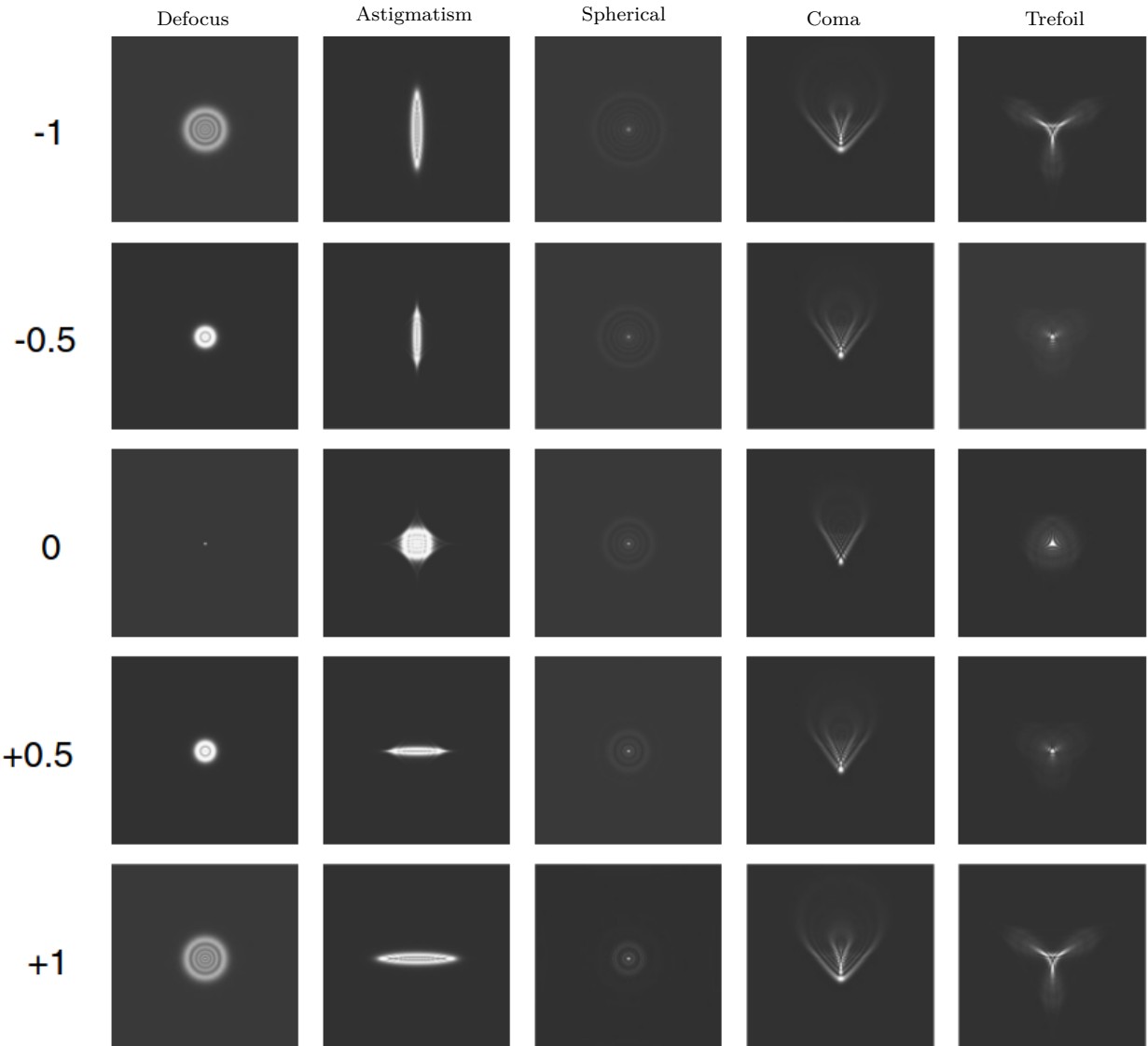

Figure 9: Different aberrations produced by single Zernike Fringe polynomials (central row) and their effect, when mixed with defocus. Figure modified from López-Gil et al. (2007). All kernels are normalized to the maximum intensity for display purposes. ALC learns a color-dependent linear combination of the corresponding Zernike polynomial to find the worst-case lens blur for a model and dataset.

dataset Russakovsky et al. (2015) has 1000 ImageNet classes and contains over 1,281k training images and 50k validation images Russakovsky et al. (2015). Additionally, we also incorporate the full ImageNet-21k dataset Russakovsky et al. (2015) to ensure a comprehensive assessment of the effectiveness of our optical attack across varying scales and complexities Russakovsky et al. (2015). To complement image classification, we also conduct experiments on image segmentation on the COCO dataset Lin et al. (2015), a comprehensive dataset designed for object detection and segmentation tasks, using SAM Kirillov et al. (2023).

## B  ALC Attack Restrictions

To restrict the ALC attack to optical boundaries, we introduce $\tau$ and $\tau_e$ as in Eqs. 7 and 8 to restrict the coefficients ($A_n^m$). In Figure 10, we demonstrate the accuracy dependency on the restriction values $\tau$ and

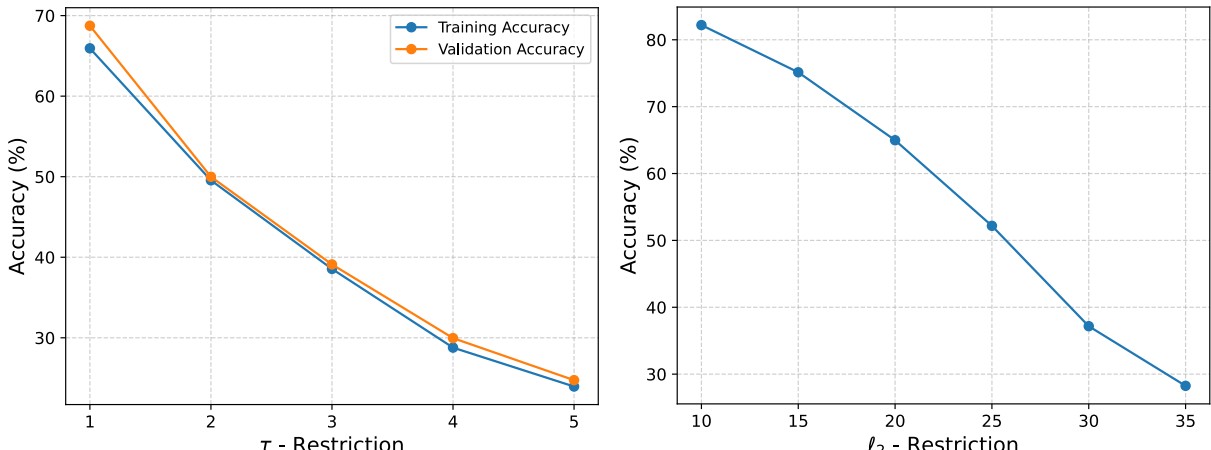

Figure 10: Ablations using the ResNet50 model on the ImageNette dataset. **Left:** Ablation study of $\tau$ and $\tau_e$ threshold from 1 to 5. A minor attack, with a low $\tau$ & $\tau_e$ threshold, only marginally lowers the accuracy of the classification model. **Right:** Ablation study of the $\ell_2$-based restriction, where we bound the $\ell_2$-distance to a maximum $\delta$ from $\delta \in [10, 35]$. A minor attack, with a low $\delta$ threshold, only marginally lowers the accuracy of the classification model.

$\tau_e$ with the ResNet50 model on the ImageNette dataset. All other experiments are conducted with $\tau = 4$ and $\tau_e = 4$ to have a similar $\ell_2$-distance as in OpticsBench Müller et al. (2023). Furthermore, to restrict the $\ell_2$-distance of the perturbed image, we integrate a $\delta$ bound and use $\nu < 1$.

## C   Model and Dataset Comparison

The initialization of ALC coefficients are the same for all experiment runs. However, over the ALC attack procedure, they start deviating from each other. The progression of the coefficients through the ALC attack is displayed in Figure 11.

Figure 12 to 14 show the variance of the ALC coefficient by multiple ALC runs. While the model architecture and training settings are constant over the five runs per model and dataset, the ALC seed is different for each run.

In Figure 15, the ALC kernels corresponding to the results in Table 2 are visualized. The first three columns represent the individual color channels, while the last column illustrates the combined ALC kernel.

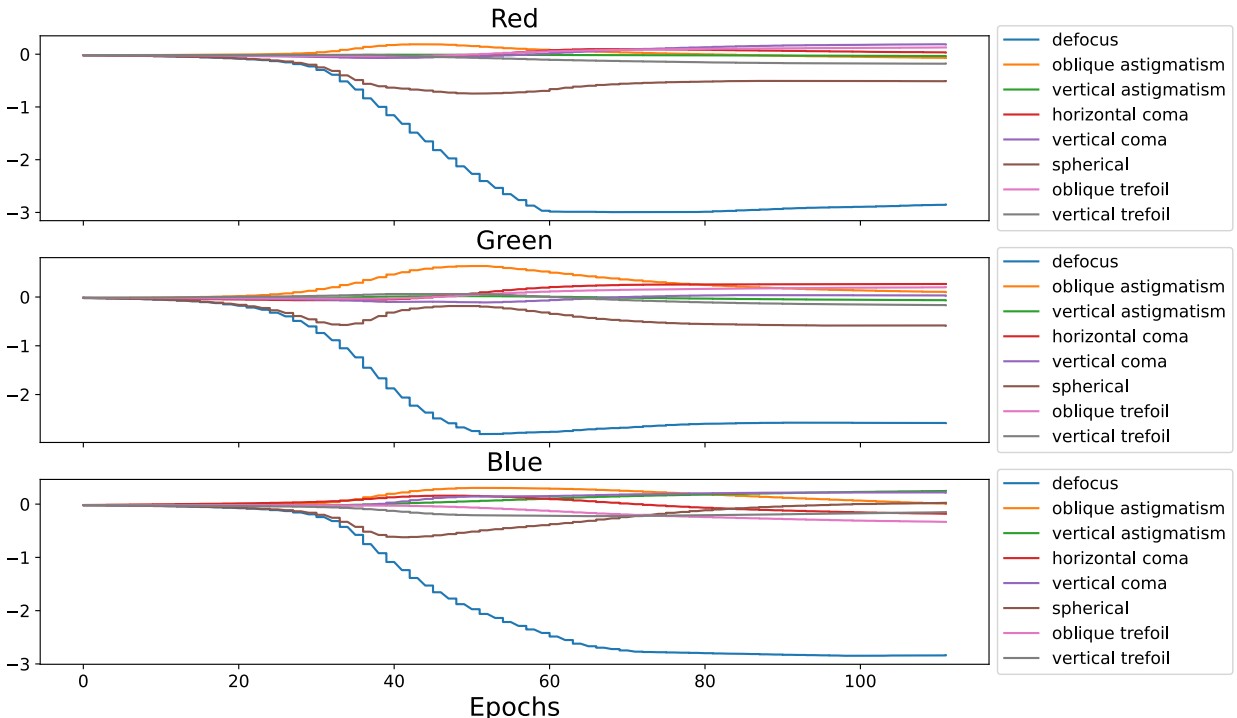

Figure 11: Updating the ALC kernels over the number of 100 epochs. The ResNet50 model He et al. (2016) is trained on ImageNette Howard (2023) and subsequently used for the ALC Attack.

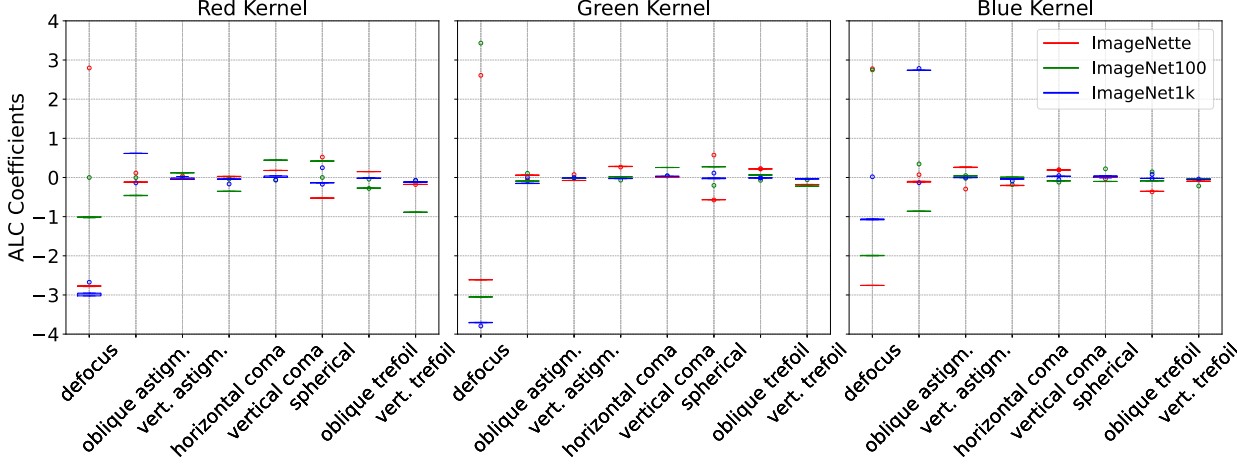

Figure 12: Comparison of the resulting ALC coefficients ResNet50 on multiple ImageNet subsets (ImageNette, ImageNet100, and ImageNet1k). The coefficient variances are low, while their respective differences are significant. Red = ImageNette, green = ImageNet100, and blue = ImageNet1k.

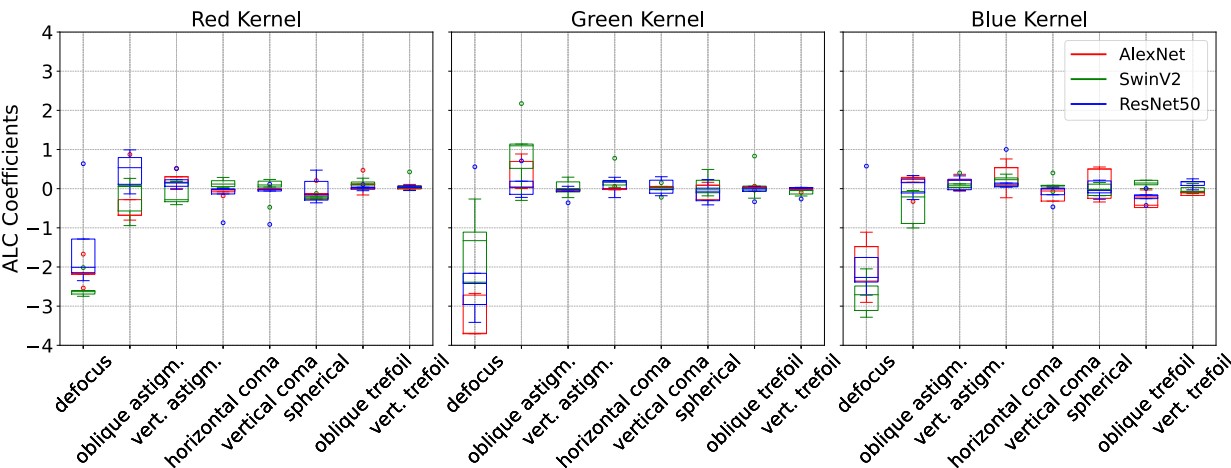

Figure 13: Comparison of the resulting ALC coefficients on multiple models (AlexNet, SwinV2, ResNet50) using ImageNette dataset. Red = AlexNet, green = SwinV2, and blue = ResNet50.

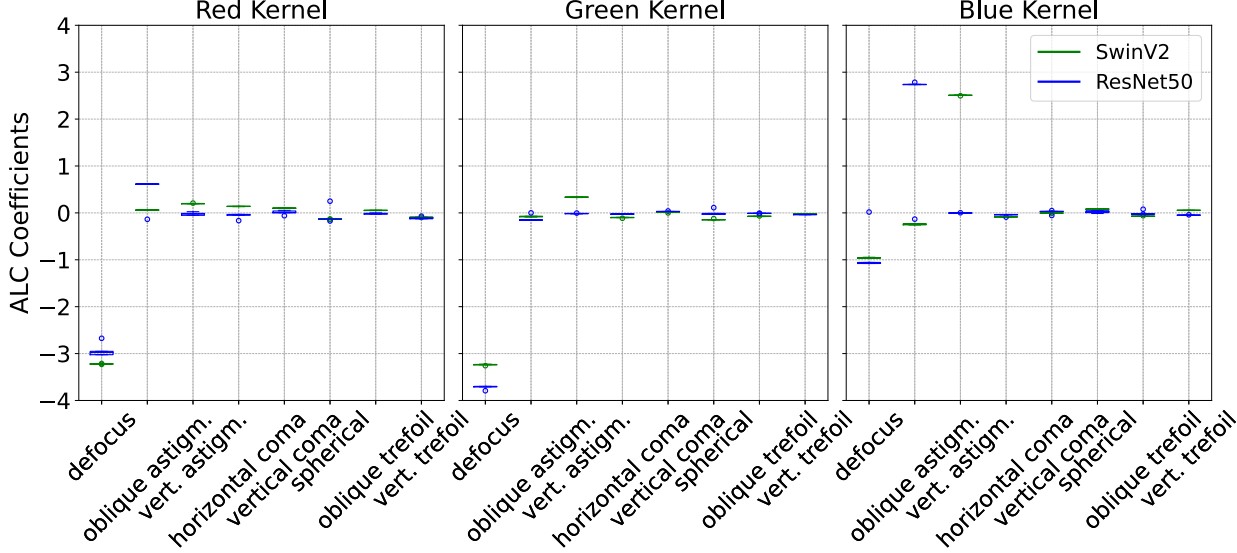

Figure 14: Comparison of the resulting ALC coefficients on multiple models (SwinV2, ResNet50) using ImageNet1k dataset. Green = SwinV2 and blue = ResNet50.

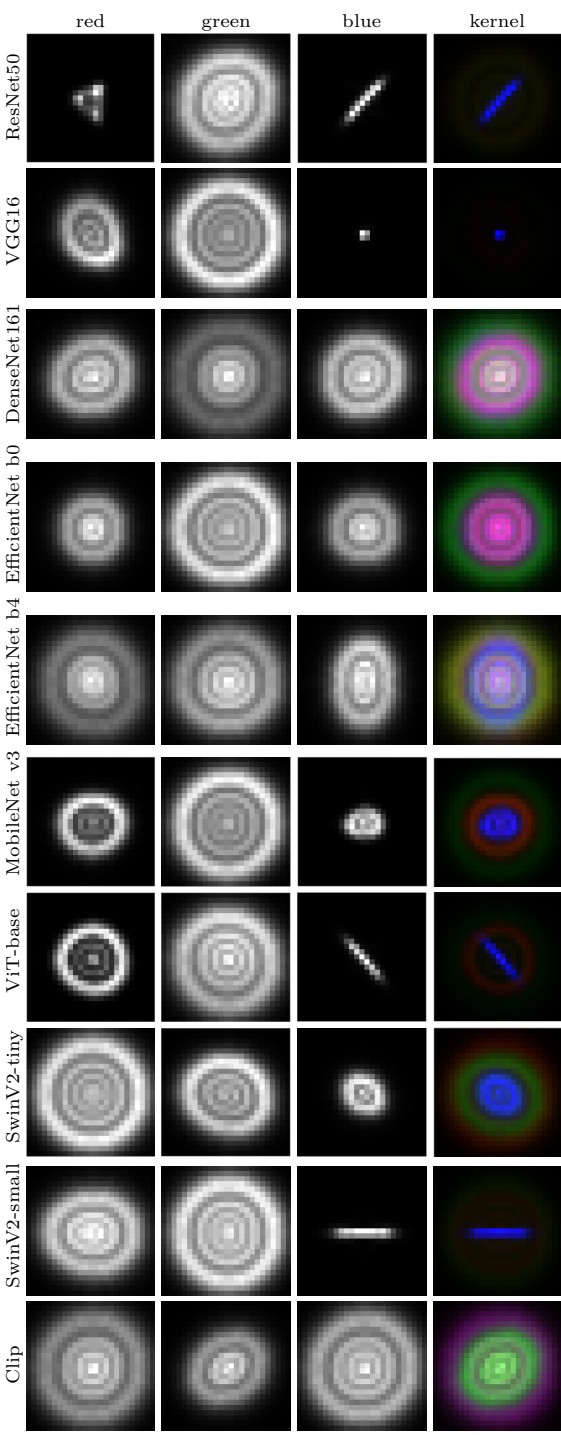

Figure 15: A comparison of the generated kernel with the ImageNet1k dataset Russakovsky et al. (2015) and all models from Table 2. The plot shows the generated kernel for the three color channels (red, green, and blue) and the combined kernel for all channels.

## D  Ablation on Optical Corruptions

In Figure 16 and 17 the corruption with the highest coefficients is removed and the history of the coefficients indicate a shift to spherical corruption in absence of defocus. Subsequently, vertical and horizontal coma are the highest corruption coefficients at the ALC attack.

Figure 18 and 19 show the variance of ALC coefficients on multiple dataset with five seeds per attack.

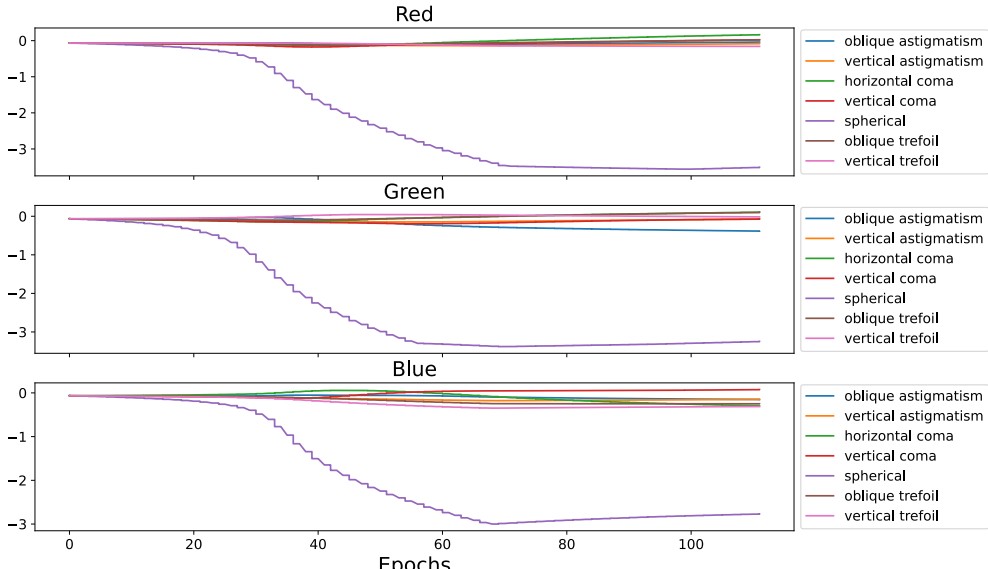

Figure 16: Running an ALC attack without the defocus optical aberration. By excluding defocus the performance decrease is lower by 4.2% in comparison to an optical attack with a defocus aberration.

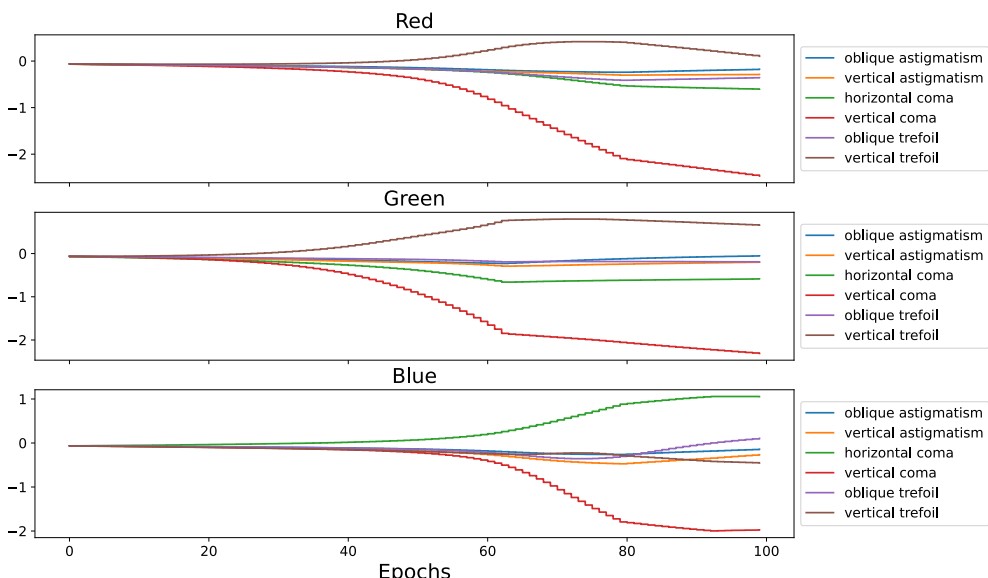

Figure 17: Running an ALC attack without the defocus and spherical optical aberration. By excluding defocus and spherical aberration, the performance decrease is 24% lower than with an optical defocus aberration.

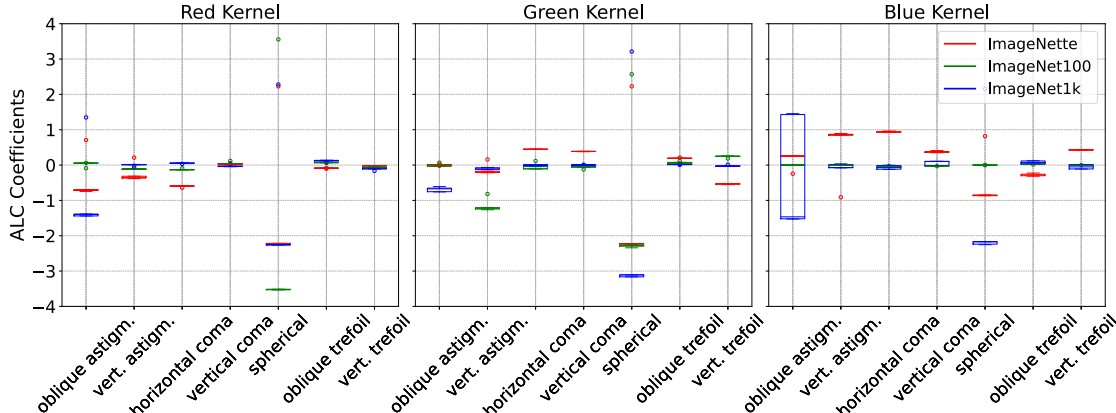

(a) Comparison of the resulting corruptions without defocus on the 3 different ImageNet subsets (ImageNette, ImageNet100 and ImageNet1k). The adversarial attack was conducted with the same model architecture (ResNet50). The models are trained with the same seed, however the adversarial attack was conducted on 5 different seeds per dataset.

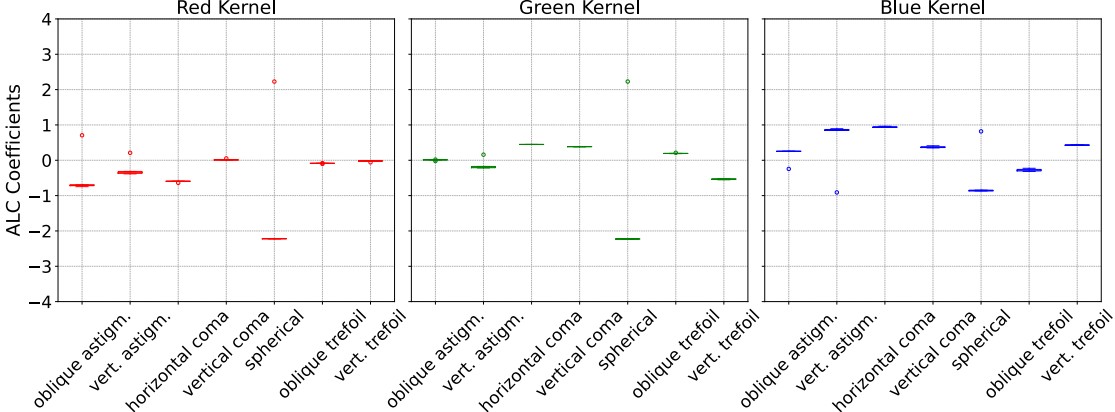

(b) Comparison of the resulting corruptions without defocus on 5 different seeds. The adversarial attack was conducted with the ResNet50 model on the ImageNette dataset. The models are trained with the same seed, however the adversarial attack was conducted on 5 different seeds.

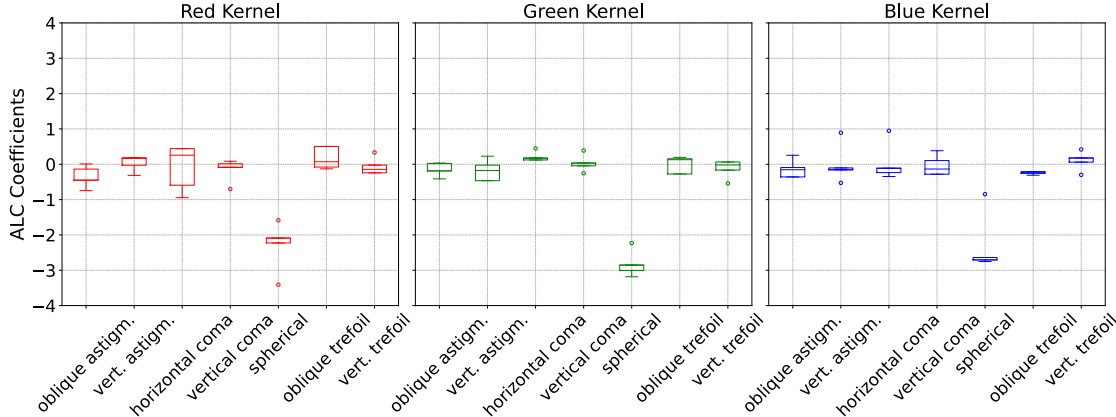

(c) Comparison of the resulting corruptions without defocus on 5 different seeds. The adversarial attack was conducted with the ResNet50 model on the ImageNette dataset. The models are trained with different seeds, and the adversarial attack was also conducted on 5 different seeds. The performance of the trained models for the multiple seeds only differ marginally (0.825, 0.816, 0.827, 0.817, and 0.806).

Figure 18: Variance of ALC coefficients on multiple dataset with five seeds per attack.

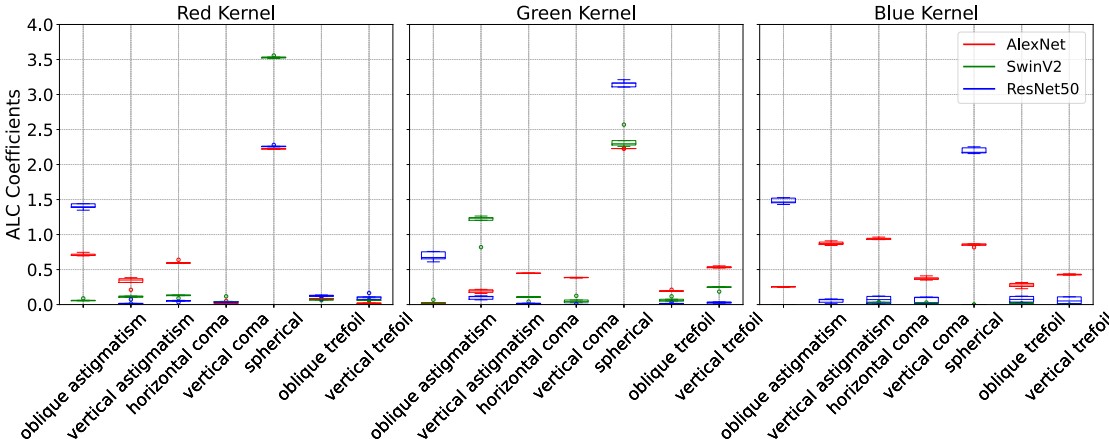

(a) Comparison of the resulting absolute corruptions values without defocus on the 3 different ImageNet subsets (ImageNette, ImageNet100 and ImageNet1k). The adversarial attack was conducted with the same model architecture (ResNet50). The models are trained with the same seed, however the adversarial attack was conducted on 5 different seeds per dataset.

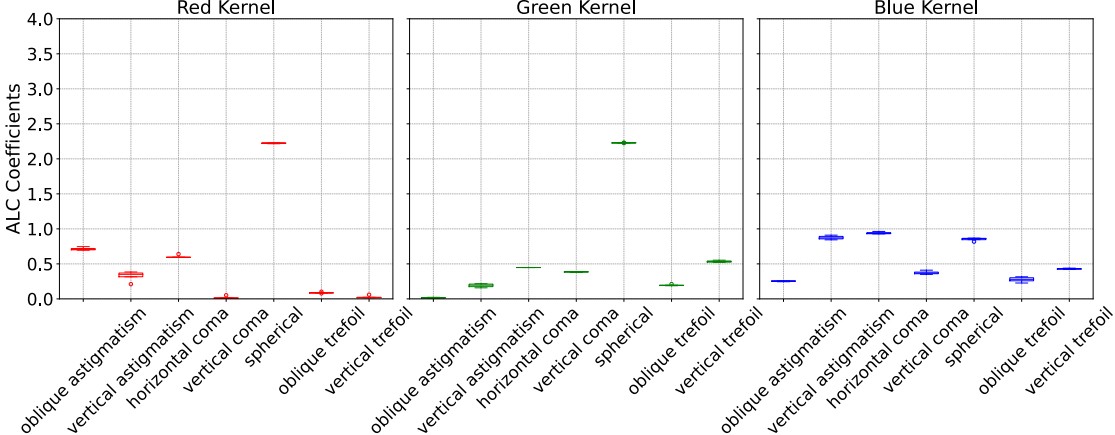

(b) Comparison of the resulting absolute corruptions values without defocus on 5 different seeds. The adversarial attack was conducted with the ResNet50 model on the ImageNette dataset. The models are trained with the same seed, however the adversarial attack was conducted on 5 different seeds.

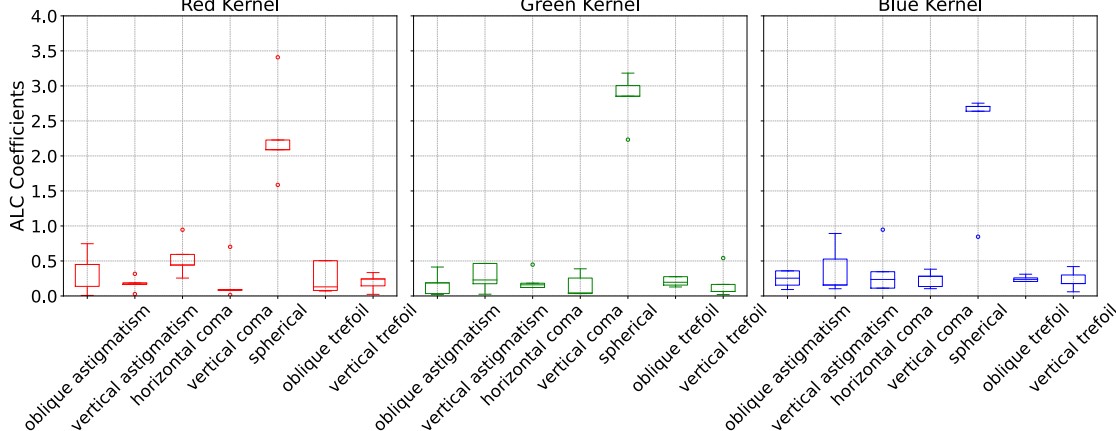

(c) Comparison of the resulting absolute corruptions values without defocus on 5 different seeds. The adversarial attack was conducted with the ResNet50 model on the ImageNette dataset. The models are trained with different seeds, and the adversarial attack was also conducted on 5 different seeds. The performance of the trained models for the multiple seeds only differ marginally (0.825, 0.816, 0.827, 0.817, and 0.806).

Figure 19: Variance of the magnitudes of ALC coefficients on multiple datasets with five seeds per attack.

# E  Image Specific Kernels

Figure 20 shows examples of image specific kernels, which are generated by the ALC attack over each image separately. The examples are randomly chosen examples from five different classes from the validation set. Furthermore, Figure 21 provides an overview of the corruption coefficient values from all ImageNette validation set images.

Table 7 shows the accuracy per class for ResNet50, SwinV2, and AlexNet. All three models have similar accuracy drops over the same classes. Furthermore, Table 8 has the ten highest (left side) and the ten lowest accuracy drops on the ImageNet1k dataset for the ResNet50 model.

|  | ResNet50 | SwinV2 | AlexNet |
|---|---|---|---|
| Tench | 0.646 | 0.729 | 0.630 |
| English springer | 0.798 | 0.883 | 0.753 |
| Cassette player | 0.241 | 0.355 | 0.361 |
| Chain saw | 0.595 | 0.627 | 0.678 |
| Church | 0.819 | 0.885 | 0.800 |
| French horn | 0.815 | 0.834 | 0.706 |
| Garbage truck | 0.815 | 0.876 | 0.777 |
| Gas pump | 0.716 | 0.763 | 0.652 |
| Golf ball | 0.306 | 0.468 | 0.197 |
| Parachute | 0.462 | 0.643 | 0.431 |

Table 7: Accuracy drop due to the per-image ALC attack, for three models (ResNet50, SwinV2, AlexNet). The attack is conducted on the ImageNette validation set.

| Class | Acc. Delta | Class | Acc. Delta |
|---|---|---|---|
| Projector | 1.000 | Espresso | 0.1867 |
| Admiral | 0.980 | CD Player | 0.2222 |
| Crash Helmet | 0.980 | Jackfruit | 0.2444 |
| Quill | 0.980 | Shopping Basket | 0.2511 |
| Bighorn | 0.978 | Trailer Truck | 0.2600 |
| Standard Poodle | 0.978 | Cairn | 0.2667 |
| Armadillo | 0.978 | Water Snake | 0.2800 |
| Table Lamp | 0.978 | Sock | 0.2911 |
| Chime | 0.978 | Tile Roof | 0.2956 |
| Prairie Chicken | 0.978 | Vestment | 0.2956 |

Table 8: Accuracy delta due to the per-image ALC attack on the validation ImageNet1k dataset with ResNet50. The ten largest accuracy delta values (left side) and the ten smallest accuracy delta (right side).

# F  Adversarial Training

While training image classification models adversarially with our proposed ALC attack, multiple hyperparameters for the adversarial process are adjustable. ALC specific hyperparameters are:

- $\tau$ & $\tau_e$ to control the intensity of the corruptions

- $\nu$ to bound the $\ell_2$-distance

- *Attack frequency*, which determines to not attack every training batch

- *Start epoch*, which sets the first epoch, in which the ALC attack will start to attack the model on the training batches

To have a stable training process, we used multiple hyperparameter combination for our experiments. The results of these experiments can be examined in Table 9.

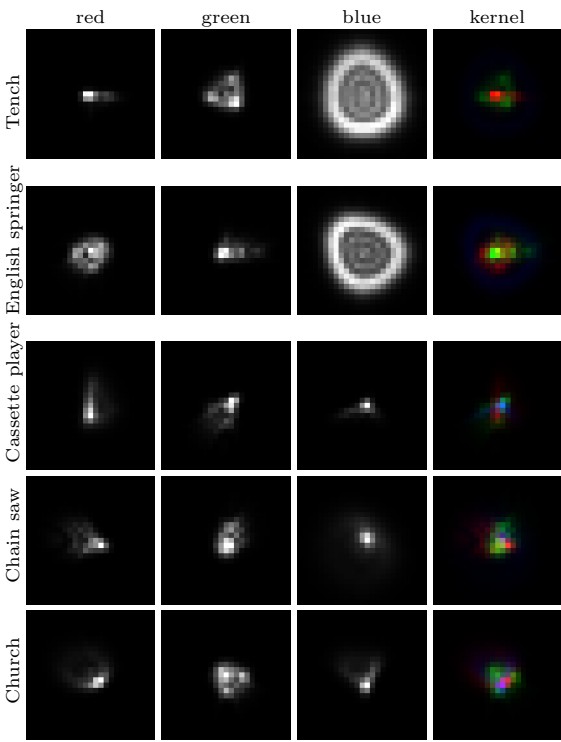

Figure 20: A comparison of the generated kernels with the ImageNette dataset Russakovsky et al. (2015) and the ResNet50 model. The plot shows generated image specific kernels for the three color channels (red, green, and blue) and the combined kernel for all channels.

| Model | $\tau$ | $AF$ | $SE$ | CD | ALC |
|---|---|---|---|---|---|
| Resnet50 | 1 | 1 | 0 | 0.781 | 0.272 |
|  | 2 | 1 | 0 | 0.770 | 0.301 |
|  | 3 | 1 | 0 | 0.772 | 0.324 |
|  | 4 | 1 | 0 | 0.761 | 0.372 |
|  | 5 | 1 | 0 | 0.713 | 0.331 |
|  | 4 | 3 | 0 | 0.761 | 0.354 |
|  | 4 | 1 | 20 | 0.755 | 0.294 |
| SwinV2 | 4 | 3 | 0 | 0.851 | 0.299 |
| AlexNet | 4 | 3 | 0 | **0.864** | **0.393** |

Table 9: Ablation of adversarial training with the ALC attack on ResNet50 He et al. (2016), SwinV2-tiny Liu et al. (2022a), and AlexNet Krizhevsky & Hinton (2009). First column section: The trained model. Second column section: The ALC hyperparameter combination with $\tau$, attack frequency ($AF$) and starting epoch ($SE$). Third column section: The result on clean data (CD) and ALC attack result.

## G  Texture bias

To measure the texture-shape bias of image classification models, we use the cue-conflict classification problem Geirhos et al. (2018). This dataset consists of 1,280 images where shape and texture cues conflict. The conflicting images are synthetically generated using a style transfer model Gatys et al. (2016) and ImageNet examples Russakovsky et al. (2015). The conflicting cues are 16 super-classes of ImageNet. From an information standpoint, it is technically correct to predict either label (or both). Nevertheless, humans prioritize shape cues for categorization, unlike most models Geirhos et al. (2018). Employing either the shape or texture cue label as the accurate classification allows for the measurement of shape accuracy and texture accuracy, respectively. According to these assessment metrics, cue accuracy is the ratio of predictions that align with either the shape or texture label to the instance of misclassification:

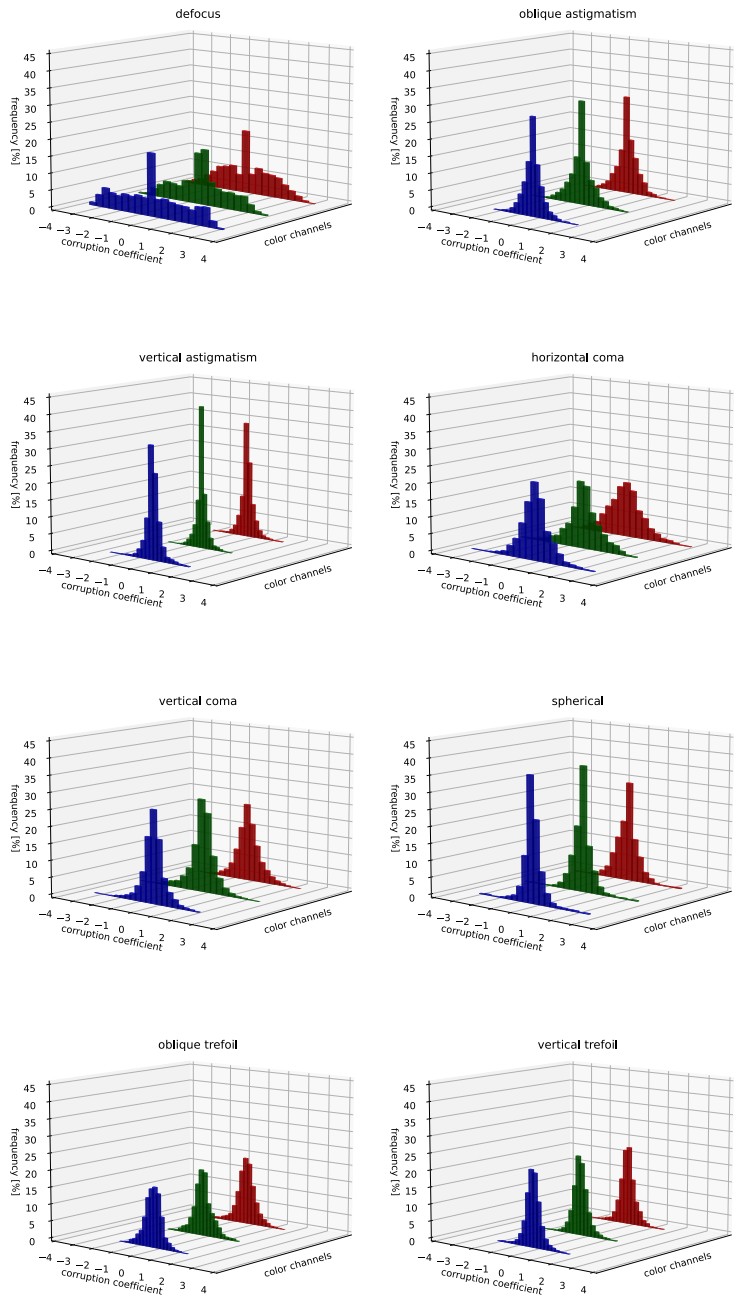

Figure 21: Histogram of corruption values ($A_n^m$) for the image specific ALC attack on the ImageNette validation set with the ResNet50.

$$Cue\ Accuracy = Shape\ Accuracy + Texture\ Accuracy \tag{9}$$

We also formalize the shape bias Geirhos et al. (2018) as the proportion of decisions made owing to shape over the number of correct decisions:

| Model | Attack | Cue Acc | Shape Acc | Texture Acc | Shape Bias |
|---|---|---|---|---|---|
| ResNet50 | - | 0.671 | 0.149 | 0.522 | 0.222 |
| ResNet50 | ALC | 0.161 | 0.112 | 0.050 | 0.684 |
| EfficientNet-b4 | - | 0.701 | 0.291 | 0.417 | 0.411 |
| EfficientNet-b4 | ALC | 0.374 | 0.311 | 0.045 | 0.868 |
| CLIP | - | 0.540 | 0.363 | 0.177 | 0.671 |
| CLIP | ALC | 0.452 | 0.365 | 0.088 | 0.868 |
| ViT-base | - | 0.682 | 0.271 | 0.411 | 0.398 |
| ViT-base | ALC | 0.338 | 0.298 | 0.039 | 0.884 |
| AlexNet | - | 0.612 | 0.161 | 0.451 | 0.264 |
| AlexNet | Adv. Train | 0.642 | 0.200 | 0.443 | 0.312 |

Table 10: Cue accuracy, shape accuracy, texture accuracy and shape bias for multiple models with and without the ALC attack. In the last row we trained an AlexNet adversarially and this against the model without the adversarial training.

$$Shape\ Bias = \frac{Shape\ Accuracy}{Cue\ Accuracy} \tag{10}$$

The evaluation of models against this metric shows that an optical attack forces the models to shift from their dependence on texture-based decisions to preferring shape-based recognition. This effect is strongly demonstrated through the shape bias of a ResNet50 model, which increases from 0.222 to 0.684. The state-of-the-art foundation model CLIP base, is known to have stronger shape bias in comparison to models, which were only trained on ImageNet1k. Comparing the CLIP shape bias from pre and post optical attack, we see a stable shape accuracy, however a drop in texture accuracy and thus also an increase in shape bias from 0.671 to 0.868. In the same way, EfficientNet-b4 and Vision Transformer models also increase shape bias from 0.411 to 0.868 and 0.398 to 0.884, respectively, but also a considerable increment in the number of correctly classified categories by shape.

Adversarial training of the AlexNet model as described in Subsection 4.7 increases the shape bias without the optical attack by 4%, while also increasing the cue accuracy by 3%. Thus, the model not only shifts its attention from texture to shape, but also is able to classify more cues correctly.

## H  Non-$\ell_2$-boundedness

In the following, we elaborate on the non-$\ell_2$-boundedness of the attack and showcase the underlying reasons. We proceed in two steps: 1) we demonstrate that the attack PSF in our approach *is* $\ell_2$-bounded in the complex amplitude, but that the square in Eq. 3 removes this property, in step 2) an argument is made that the convolution itself also results in unbounded pixel differences, regardless of the PSF properties.

Regarding step 1), we note that the Zernike basis is orthonormal in the Fourier space of the lens, as is its Fourier transform Dai (2006) in the spatial domain. We can therefore consider Parseval's theorem to hold in the complex amplitude (the wave-optical PSF). Computing the intensity PSF from the wave-optical complex amplitude requires taking a square which removes the linearity. As for step 2), consider a PSF that is a shifted Dirac pulse: convolution with this kernel shifts an image by a discrete amount. The pixel-wise differences are therefore unbounded.

To conclude: it may be possible to $\ell_2$-bound ALC by working in the complex amplitude domain and removing the possibility of image shifts. We have not done so because it relates to the unpractical assumption of coherent illumination, which is usually only available in laboratory settings.

## I  Additional Practical Aspects of Lens Aberrations

In this study, we have limited the Zernike modes to a subset of eight different lower-order modes. From a physical perspective, these primary aberrations are the dominant contributors to the overall amount of aberrations. Further, in optical design, these are crucial to compensate and only small residual aberrations left are of higher order Gross, Herbert (2006). Restricting our attack to primary aberrations serves two

purposes: physically, it reflects general lens design constraints; from a machine learning perspective, it acts as a regularization of the search space, ensuring that the identified vulnerabilities correspond to structural optical failures rather than high-frequency adversarial noise.

While we can identify worst-case aberration configurations, the obtained lens blur represents a worst-case point in the continuous Zernike space, which may not directly correspond to a specific manufacturing flaw of a single lens element. This can be addressed by more carefully constraining the parameter space and search directions. Therefore, ALC should be viewed as a probing tool for *optical tolerances* and model sensitivity, rather than a direct simulation of a specific lens manufacturing defect.

Although most of the aberrations are covered by the lens design and tolerances, the manufacturing process itself can introduce aberrations of higher order[1]. To illustrate this, we give a brief example. A synchro-speed polishing machine leads to a linear combination of Fringe modes $A_n = \{9, 16, 25\}$, which introduces locally high gradients in the pupil phase. The lens blur of such a lens is more irregular and stronger compared to the expected lower-order blur from the lens design. This holds true when both lens blurs would have the same RMS wave aberration error. Therefore, the interpretation of our results with ALC should be guided with further testing of real lenses of typical effects. Including the manufacturing process in our simulation is a promising future research direction, which could *e.g.* provide sensitivity maps that inform manufacturing tolerances of which manufacturing process to use for a given image classification model.

However, in the current definition, ALC is not integrated into an end-to-end or co-optimization process of an actual lens design. Without this, it may still be hard to directly make use of the results obtained by ALC for a specific lens design. ALC therefore offers only a first step into a promising method of deep optics, and we are open for improvements and collaboration, such as testing the integration into an actual lens design process.

---

[1]This detail was, among others, confirmed in a personal communication with an expert optic designer.