# OpenReview forum: "Parameterized Adverse Lens Corruptions to Probe Model Robustness to Optical Tolerances"
_TMLR — Accepted by TMLR_

### Review · Reviewer_3dFg · 2026-03-10

**Summary Of Contributions:**

The paper proposes ALC, an adversarial optical-blur attack that optimizes Zernike-based aberration coefficients to find worst-case lens corruptions for a given model and dataset. Its main contribution is a physically grounded and interpretable corruption space for robustness analysis, together with experiments across classification and segmentation models and an initial study of ALC-based adversarial training.

**Audience:**

Yes

**Audience Explanation:**

1) At least part of the TMLR audience would likely be interested in this paper because it studies a meaningful and relatively underexplored robustness setting: physically structured optical blur attacks rather than standard pixel-space perturbations.

2) The paper’s core idea, optimizing Zernike-based lens aberration coefficients to generate model- and dataset-specific worst-case blur, is relevant to readers working on robustness, adversarial ML, distribution shift, and vision systems deployed in real imaging pipelines.

3) The paper is also likely to interest researchers because it goes beyond a narrow attack demo and evaluates the method across several datasets and model families, including ImageNet variants, CLIP, DINOv2, and SAM, and also explores adversarial training with the proposed corruption model.

4) That breadth makes the findings potentially useful both for people interested in robustness evaluation and for those interested in training-time hardening against structured corruptions.

**Broader Impact Concerns:**

Limited but non-zero concerns. Because the paper develops a physically grounded adversarial optical attack that can deliberately degrade vision models through lens-aberration-based blur, it has some dual-use risk for attacking or stress-testing real imaging systems; a brief Broader Impact statement should acknowledge this misuse potential and emphasize the intended defensive use for robustness evaluation and training.

**Claims And Evidence:**

Yes

**Claims Explanation:**

The submission provides meaningful empirical evidence that ALC can degrade performance: the paper evaluates many classification models on ImageNet-1k, reports lower accuracy than OpticsBench at roughly comparable distortion magnitude in Table 2, shows dataset/model specificity in Tables 3–5, and extends the analysis to SAM segmentation and adversarial training. These results support the narrower claim that the proposed optimization can find strong blur corruptions in the authors’ simulated setup.

However, the evidence is not fully convincing or clear enough to support the paper’s broader claims.

- First, the main evaluation often uses one optimized kernel per model–dataset pair, which is efficient for benchmarking but is weaker than a standard per-image adversarial threat model; the stronger image-specific version is only treated later and more briefly.
- Second, the PGD comparison is explicitly not directly comparable because the attacks use different constraint sets, so it only weakly supports claims of complementarity.
- Third, the paper makes strong claims about physical plausibility and “optical tolerances,” but the main paper only shows simulated PSF-based experiments and does not validate that the learned worst-case kernels correspond to realistic real-lens failures or camera measurements.

**Requested Changes:**

1) The paper positions ALC as a physically grounded optical adversarial attack, but the empirical comparisons are mainly against OpticsBench and PGD. Since the paper itself cites prior optical attack work, including Optical Adversarial Attack, the authors should provide a clearer discussion of what is genuinely new here and, ideally, include a direct experimental comparison to the closest prior optical/physical attack baselines. This is important to establish the paper’s methodological contribution beyond “a stronger benchmark corruption.”

2) As submitted, the core equations are difficult to read due to malformed rendering and awkward references such as “Eq. equation 1/3/5.” Since the contribution depends on the exact optical model, optimization, and constraints, the method section needs careful cleanup so that symbols, variables, and equation references are unambiguous and reproducible from the main paper.

3) A large part of the main evaluation uses a single optimized kernel per model–dataset pair, while the image-specific version is introduced later as a separate experiment. The paper should clearly distinguish between:
(a) a dataset-level worst-case corruption used as a robustness probe/benchmark, and
(b) an image-specific adversarial attack.
Right now, these two settings are somewhat blended, which weakens the interpretation of the main claims.

4) The paper repeatedly defers important details to Appendix A.1, F, G, and I. For a methods paper, the main text should contain enough detail to understand and reproduce the optimization procedure, attack hyperparameters, training protocol, and practical constraints without needing to reconstruct the method from appendices.

---

### Review · Reviewer_cbuh · 2026-03-15

**Summary Of Contributions:**

Authors of this paper introduce adverse lens corruption (ALC) designed to evaluate the robustness of image classification models against optical blur corruptions. ALC focuses on lens blur but complements to standard noise-based adversarial attack. Extensive experiments demonstrate that ALC not only is useful as an adversarial white-box attack, but also increases model robustness.

**Audience:**

Yes

**Audience Explanation:**

Adversarial attack and adversarial training of models are important for modern AI. The proposed work focuses on lens blur, but it could be extensively used for various scenarios to increase model robustness. Another key point is that ALC can augment with standard noise-based attacks can achieve improved performance.

**Broader Impact Concerns:**

I do not have any concerns on the ethical implication of the work.

**Claims And Evidence:**

Yes

**Claims Explanation:**

This work studied optics-driven blur-based adversarial attacks. It is different from standard noise-based adversarial attacks. Authors proposed ALC which is optimized to find the worse-case lens blur per datasets and model using gradient ascent on the linear combination of aberration basis vectors. Extensive experiments demonstrated that ALC can be useful as an adversarial white-box attack and for adversarial training of models to be robust against lens blur. Another important observation is that ALC is complementary to standard noise-based adversarial attacks.

**Requested Changes:**

In section 3, the proposed ALC is built on Goodman (2017) with several practical simplifications to the general setting. It would be better to provide more intuitive explanation of (3) and (4) on how they are related to lens blur.

Eq. equation seems duplicated works. Either Eq. or equation should be good.

---

### Review · Reviewer_6wzA · 2026-03-30

**Summary Of Contributions:**

This paper presents an image editing procedure that simulates the image formation effects of imperfect lenses. The method is intended both as a research tool for studying the challenges that optical distortions introduce to natural image classification systems and as a foundation for developing mitigation strategies. Specifically, the authors cast optical distortion as an adversarial attack problem. The proposed adverse lens corruption (ALC) procedure optimizes the coefficients of Zernike polynomials to construct an incoherent point spread function (iPSF), which is then convolved with clean images to generate attacked examples. The resulting distorted test images are shown to degrade performance across several widely used network architectures on ImageNet classification and COCO segmentation benchmarks. The paper further shows that adversarial training with ALC-edited images can modestly improve robustness to optical distortions.

**Audience:**

Yes

**Audience Explanation:**

Robustness is a critical topic within ML community and should be interesting to TMLR audience.

**Broader Impact Concerns:**

Not relevant

**Claims And Evidence:**

No

**Claims Explanation:**

My main concern with this manuscript is the positioning of the work with respect to the adversarial attack literature versus the optical distortion literature. If ALC is intended to be presented as a new adversarial attack, the evaluation currently appears somewhat incomplete, as it does not include comparisons with established attack methods such as PGD[1], DeepFool [2], CW Attack [3], or AutoAttack [4], nor does it clarify how the method relates to commonly used robustness benchmarks [5].

On the other hand, if the primary goal is to introduce ALC as a tool for studying the impact of imperfect lenses on model performance, I found the current empirical evidence insufficient to fully establish that the generated images faithfully resemble naturally occurring optical distortions. Because adversarially generated pattern could be an over-kill or mis-representation of real distortion we would see in natural environment.  In addition, the manuscript does not yet demonstrate whether the proposed approach can improve model performance on original real-world datasets that naturally contain distortions introduced by imperfect imaging systems. Showing such benefits would considerably strengthen the practical significance of the work. More broadly, support for this claim would likely require evaluation on more than a single dataset, ideally across multiple datasets that are representative of naturally occurring optical distortions.

[1]. Towards Deep Learning Models Resistant to Adversarial Attacks
[2]. DeepFool: a simple and accurate method to fool deep neural networks
[3]. Towards Evaluating the Robustness of Neural Networks
[4]. Reliable Evaluation of Adversarial Robustness with an Ensemble of Diverse Parameter-free Attacks
[5]. https://robustbench.github.io/

**Requested Changes:**

1. If the authors intend to position this work as a new adversarial attack method, they should include the standard experimental comparisons expected in the adversarial robustness literature, as noted in my summary. In particular, the paper would benefit from comparisons against established attack methods and from clarifying the method’s standing relative to commonly used robustness benchmarks such as RobustBench. This would help readers and reviewers better assess the novelty and significance of the proposed approach.

2. If the authors instead intend to position this work as a research tool for studying optical distortion, they should provide stronger evidence that the generated images are representative of distortions encountered in real-world imaging conditions. Beyond the current experiments, I would also expect the proposed procedure to be evaluated as a data augmentation method, with the aim of improving model performance on original datasets that naturally contain imperfections introduced by real imaging systems. Such claims would be substantially more convincing if they were validated on multiple representative datasets—for example, GTSRB [1], Food-101 [2], subset of these dataset that is representative or other natural imaging benchmarks—where the authors can clearly demonstrate performance gains and attribute those gains specifically to improved robustness against imperfect lens imaging.

[1]. https://benchmark.ini.rub.de/gtsrb_dataset.html
[2]. https://data.vision.ee.ethz.ch/cvl/datasets_extra/food-101/

---

### Decision · Action_Editor_X7Rg · 2026-06-12

**Recommendation:** Accept with minor revision

**Additional Comments:**

I will recommend the authors to consider reviewers' comments on clarify the positions of this paper.

More specifically, (1) state consistently whether ALC is a dataset-level robustness probe, an image-specific attack, or both, and scope each claim accordingly (6wzA, 3dFg); (2) either soften the "real optical tolerances" framing or add evidence linking the learned kernels to real lens behavior (6wzA, 3dFg); (3) clarify the PGD comparison's differing constraint sets and temper complementarity claims (3dFg); (4) fix the malformed equation references and add intuition for Eqs. (3)–(4) (cbuh, 3dFg); (5) add a brief dual-use Broader Impact statement (3dFg).

**Audience:**

Yes

**Audience Explanation:**

clearly relevant.

**Claims And Evidence:**

Yes

**Claims Explanation:**

Two of three reviewers find the evidence sufficient (cbuh recommends Featured Certification; 3dFg confirms the core results). 6wzA marks claims as "No," but the concern is about positioning and scope, not the correctness of the reported numbers. No reviewer disputes the measured results. The gap is that the paper alternates between framing ALC as an adversarial attack and as a tool for studying real optical distortion, and the "physical plausibility" claims exceed the simulated-only evidence. These are revision matters, so I judge the appropriately-scoped claims supported.

---

> ### Author Response · Authors · 2026-07-06
> **Summary of Revisions in Response to the Decision Letter - I**
>
> We thank the Action Editor and the three reviewers for their careful assessment of our work and for the constructive feedback. We have revised the submission to address all five points raised in the decision letter. In the following, we restate each point, explain the reasoning behind our revision, and describe the corresponding changes in the paper. Unless stated otherwise, all section, table, figure, and equation numbers refer to the revised version.
>
> ______
> ## 1. Consistent scoping of ALC as a dataset-level robustness probe and an image-specific attack (Reviewers 6wzA, 3dFg)
>
> The reviewers correctly observed that the original paper alternated between presenting ALC as an adversarial attack and as a tool for studying optical distortion, and that the dataset-level and image-specific settings were insufficiently distinguished. We agree that this blending weakened the interpretation of the main claims, and we have restructured the framing of the paper around two explicitly defined modes of operation.
>
> First, the abstract and the first contribution in Section 1 now introduce ALC as an optics-driven robustness probe that identifies worst-case lens blurs through adversarial optimization. The contribution statement explicitly announces the two modes: a generic mode that produces a single kernel per model and dataset, used in Sections 4.1 through 4.5, and an image-specific mode, used in Section 4.6.
>
> Second, Section 4 opens with a new paragraph that motivates the order in which the two modes are introduced. The evaluation proceeds from the probe setting, through the attack setting, to the defense setting, and it begins with the generic mode for a physical reason: a real lens resides in a single, fixed aberration state that affects all captured images alike. The image-specific mode, in contrast, no longer corresponds to a single physical lens state; instead, it provides an upper bound on the achievable optical degradation and reveals which images and classes are particularly vulnerable. Section 4.7 closes the loop by using the corruptions identified by ALC for adversarial training. Each subsection of Section 4 is now explicitly assigned to one of the two modes in this opening paragraph.
>
> Third, a new paragraph entitled "Implementation of the two modes" in Section 4 describes the optimization procedure of both modes in the main text. This change also responds to Reviewer 3dFg's remark that important methodological details were previously deferred to the appendices.
>
> Finally, the terminology within the experimental sections was made consistent with this scoping.
> ______
> ## 2. Framing with respect to real optical tolerances (Reviewers 6wzA, 3dFg)
>
> The reviewers noted that the claims regarding physical plausibility exceeded the simulation-only evidence. Of the two options offered in the decision letter, we chose to soften the framing. The revised paper therefore scopes all claims to what the simulation supports and states the remaining limitation explicitly.
>
> Concretely, we made the following wording changes. In the abstract and in Sections 1 and 2, we softened statements such as "adversarial blur in a realistic way" to formulations like "adversarial blur inspired by real optical aberrations".
>
> In addition, Section 2 contains a new passage that positions ALC relative to prior optical and physical attacks. This passage also clarifies what is methodologically new relative to the prior optical-attack literature cited in the paper, a point raised by Reviewer 3dFg.
>
> Furthermore, in Section 5 the earlier claim that our constraints "enforce physical plausibility, meaning the identified vulnerabilities correspond to potential real-world optical degradations" was removed. The revised text states that the constraints restrict the search to the set of feasible optical aberrations rather than allowing arbitrary pixel manipulation. It also points out that a worst-case point in the continuous Zernike space is not guaranteed to correspond to a single manufacturable lens.
> ______
> ## 3. PGD comparison and complementarity claims (Reviewer 3dFg)
>
> We agree that the comparison between ALC and PGD must not be read as a comparison of attack strength, since the two methods operate under different constraint sets. The revised paper makes the nature and the limits of this comparison explicit in three places.
>
> In Section 1, the complementarity claim now names the differing constraint set. In Section 4.6, the paragraph describing the experiment now states that ALC is constrained in Zernike-coefficient space rather than by an $\ell_p$ budget. In Section 5, a new discussion paragraph places ALC in the context of standardized evaluation practice. The paragraph concludes that ALC should be viewed as a complementary diagnostic tool and training regularizer rather than as a direct competitor to pixel-budget attacks.

---

> > ### Author Response · Authors · 2026-07-06
> > **Summary of Revisions in Response to the Decision Letter - II**
> >
> > ## 4. Malformed equation references and intuition for Equations (3) and (4) (Reviewers cbuh, 3dFg)
> >
> > We corrected all occurrences of the malformed references in Section 3 and Appendix B, so that equation references now render uniformly as "Eq. X". To make the optical model accessible to readers without a background in Fourier optics, we added explanatory text at three places in Section 3.
> > ____
> > ## 5. Broader Impact Statement (Reviewer 3dFg)
> >
> > We added a Broader Impact Statement after the Conclusion. The statement acknowledges that ALC demonstrates a method to intentionally degrade model performance via optical blur and that this carries a dual-use potential for stress-testing real imaging pipelines.
> > ____
> > ## Further changes
> >
> > Beyond the five points above, we corrected several issues that we identified during the revision. We list them here for completeness and transparency.
> >
> >  - Corrected the overview list at the beginning of the supplementary material.
> >  - Unified inconsistent numbering styles.
> >  - Realigned several figures.
> >  - Fixed various typos.
> >
> > We hope that the revised version addresses all concerns raised in the decision letter.